# A Dynamic Model of Performative Human-ML Collaboration: Theory and Empirical Evidence

## Abstract

Machine learning (ML) models are increasingly used in various applications, from recommendation systems in e-commerce to diagnosis prediction in healthcare. In this paper, we present a novel dynamic framework for thinking about the deployment of ML models in a performative, human-ML collaborative system. In our framework, the introduction of ML recommendations changes the data-generating process of human decisions, which are only a proxy to the ground truth and which are then used to train future versions of the model. We show that this dynamic process in principle can converge to different stable points, i.e. where the ML model and the Human+ML system have the same performance. Some of these stable points are suboptimal with respect to the actual ground truth. As a proof of concept, we conduct an empirical user study with 1,408 participants. In the study, humans solve instances of the knapsack problem with the help of machine learning predictions of varying performance. This is an ideal setting because we can identify the actual ground truth, and evaluate the performance of human decisions supported by ML recommendations. We find that for many levels of ML performance, humans can improve upon the ML predictions. We also find that the improvement could be even higher if humans rationally followed the ML recommendations. Finally, we test whether monetary incentives can increase the quality of human decisions, but we fail to find any positive effect. Using our empirical data to approximate our collaborative system suggests that the learning process would dynamically reach an equilibrium performance that is around 92% of the maximum knapsack value. Our results have practical implications for the deployment of ML models in contexts where human decisions may deviate from the indisputable ground truth.

## 1 Introduction

Human-ML collaboration is increasingly used in various applications, from content moderation in social media (Lai et al., 2022) to predicting diagnoses in healthcare (Jacobs et al., 2021; Dvijotham et al., 2023) and making hiring decisions in human resources (Peng et al., 2022). Companies that implement human-ML collaborative systems face three crucial challenges: 1) ML models learn from past human decisions, which are often only an approximation to the ground truth (noisy labels); 2) ML models are rolled out to help future human decisions, affecting the data-generating process of human-ML collaboration that then influences future updates to ML models (performative predictions as in Perdomo et al. (2020)); and 3) the quality of the human-ML collaborative prediction of the ground truth may change as a function of incentives and other human factors. These challenges create a dynamic learning process. Without access to the ground truth, it is often difficult to know whether the learning process will reach an equilibrium state with a good approximation of the ground truth, if it is interrupted at a sub-optimal level, or if it does not reach a stable state at all.

For intuition, we can focus on the decision of a healthcare company to develop and deploy an ML model to predict medical diagnoses from patient visits. The problem is made difficult by the fact that a doctor's diagnoses can be wrong, and it is often too costly or time-consuming to identify the indisputable ground truth—i.e., the underlying true diagnosis of a patient—so the company typically uses all diagnoses to train their ML model, without distinction between good or bad diagnoses. In addition, the company typically evaluates the algorithm's performance based on its ability to match

those same doctor diagnoses, potentially replicating their mistakes. The dynamic deployment of updates to ML models that support doctor diagnoses could lead to a downward spiral of human+ML performance if the company deploys a bad model and the bad model adversely affects doctor decisions. Or, it can lead to continuous improvement until it reaches a stable point that is a good approximation to the indisputable ground truth. Without (potentially costly) efforts to measure the ground truth, the company has no way of distinguishing between downward spirals or continuous improvements.

This raises a multitude of empirical questions regarding the governing mechanisms of this dynamic system. How do humans improve on ML predictions of different quality levels, and do financial incentives matter? Will the dynamic learning process converge to a good equilibrium even without the company knowing the actual ground truth labels?

**Contributions.** In this paper, we present a novel framework for thinking about ML deployment strategies in a performative, human-AI collaborative system. We present a theoretical framework to identify conditions under which ML deployment strategies converge to stable points that are a good approximation to the ground truth, and conditions under which there are downward spirals away from the ground truth. Our theory introduces the notion of a collaborative characteristic function, which maps algorithmic performance to the performance of human decisions supported by ML predictions. As a proof of concept for our theory, we provide an empirical study in which humans solve knapsack problems with the help of machine learning predictions. We conducted a user study with 1,408 participants, each of whom solved 10 knapsack problems. The empirical exercise allows us to evaluate 1) the quality of human decisions supported by ML models of varying performance, 2) how these human decisions compare to a best-case scenario, and 3) how these human decisions are affected by monetary incentives. With some additional assumptions, we can map these data to our theoretical collaborative characteristic function.

We highlight three main empirical results. First, we show that humans tend to improve upon the ML recommendation for many levels of ML performance. Second, humans sometimes submit solutions that are worse than the ML recommendation, despite the fact that with knapsack, it is fairly easy for them to compare their solution to the ML suggestion and pick the best of the two. Third, humans do not respond to financial incentives for performance. The empirical data can be used to approximate theoretical collaborative characteristic functions. The results suggest that, at least in our context, collaborative characteristic functions are invariant to monetary incentives. Additionally, the fact that humans sometimes submit solutions that are worse than the provided ML recommendation implies that there remains a gap between the collaborative characteristic function based on human labels and the collaborative characteristic function constructed by selecting the maximum between the human and ML solution.

Our results have practical implications for the deployment of ML models when humans are influenced by those models but their decisions deviate from an unknown ground truth. First, performance metrics of ML models can be misleading when the learning objective is based on comparisons against human decisions and those decisions can be wrong. Companies should thus exert efforts to assess the quality of human decisions and take that into account when training ML models. For example, in the medical setting, human diagnoses should be first verified or confirmed by external experts, or patients should be followed up to confirm the validity of initial diagnoses. At a minimum, ML models should be trained on subsets of data for which there is enough confidence that the decisions are correct. Second, our work highlights the strategic importance of deploying ML models that allow for convergence to a stable point with higher utility than humans alone. Such convergence is not guaranteed and, as argued above, difficult to assess. Third, our work calls for the need to adopt a dynamic approach when deploying algorithms that interact with human decisions, and those interactions are used for future model building.

## 2 RELATED WORK

There has been a growing body of work investigating various forms of **human-ML collaboration**. From learning-to-defer systems, where a model defers prediction tasks to humans if its own uncertainty is too high (Cortes et al., 2016; Charusaie et al., 2022; Mozannar et al., 2023), to ML-assisted decision making where humans may or may not consult ML predictions to make a decision (Mozannar et al., 2024c; Dvijotham et al., 2023; Jacobs et al., 2021). Several alternative decision mechanisms have also been explored (Steyvers et al., 2022; Mozannar et al., 2024a). The application areas

range from programming (Dakhel et al., 2023; Mozannar et al., 2024b), to healthcare (Jacobs et al., 2021; Dvijotham et al., 2023) and business consulting (Dell'Acqua et al., 2023). Related work also investigates factors influencing human-ML collaboration, such as explanations of ML predictions (Vasconcelos et al., 2023), monetary incentives (Agarwal et al., 2023), fairness constraints (Sühr et al., 2021), and humans' adaptability to model changes (Bansal et al., 2019). In this work, for the first time to the best of our knowledge, we theoretically examine the human+ML interaction from a dynamic perspective, where ML models learn from human decisions that are 1) the result of previous human+ML collaboration and 2) can arbitrarily deviate from the underlying ground truth.

This paper is also inspired by an extensive line of work on **performative prediction** (Perdomo et al., 2020; Mendler-Dünner et al., 2020; Hardt et al., 2022; Mendler-Dünner et al., 2022), a theoretical framework in which predictions influence the outcome they intend to predict. We adapt the ideas of performative prediction to a context of human-ML collaboration and extend it in three major ways: 1) In our setting, the model predictions change the quality of the human-ML labels as a proxy for the ground truth (e.g., a doctor diagnosis), but the ground truth is held constant (e.g., the true patient diagnosis); 2) We introduce the concept of utility, to quantify the quality of a solution with respect to the ground truth. There can be several stable points with respect to model parameters in the performative prediction framework, but not all of them have the same utility, i.e., are equally good at approximating the indisputable ground truth; 3) The ground truth is unknown, and the mapping between human or ML labels and the ground truth is not fixed. To the best of our knowledge, we are the first to explore performative predictions where the deployment of ML models occurs while the model's performance relative to the ground truth is unknown, and only its similarity to human labels is available. Our empirical application is also novel in that it provides a first step towards investigating the implications of performative predictions for human-ML collaboration.

## 3 PROBLEM STATEMENT

We consider a setting in which time is separable in discrete time epochs $t = 1, ..., T$. At each $t$, a firm deploys machine learning model $M_t \in \mathcal{M}$ of a model class $\mathcal{M}$, with $M_t : \mathcal{X} \to \mathcal{Y}$. The model $M_t$ predicts a solution $Y \in \mathcal{Y}$ (e.g., a diagnosis) to a problem $X \in \mathcal{X}$ (e.g. the patient's symptoms) as a function of past data. The firm employs expert humans $H \in \mathcal{H}$ with $H : \mathcal{X} \times \mathcal{Y} \to \mathcal{Y}$, who solve the problems with the help of ML predictions. We will write $M_t(X) = Y_{M_t}$ and $H(X, Y_{M_t}) = Y_{H_t}$. We assume that for all $X \in \mathcal{X}$, there exists an optimal solution $Y^*$, which is the indisputable ground truth.

**The Firm's Learning Objective.** In many real-world applications, determining the ground truth label $Y^*$ can be extremely costly. For example, obtaining the correct medical diagnosis can often require the knowledge of various specialists (e.g., orthopedists, pediatricians, neurologists). Even when a single expert is enough, they can misdiagnose a patient's symptoms. Yet, in many of these cases, using the human labels $Y_{H_t}$ as a proxy for $Y^*$ is the only feasible option to build ML models. We allow the quality of $Y_{H_t}$ with respect to $Y^*$ to change. This means that two iterations of the ML model, $M_t$ and $M_{t+1}$, are trained on data from two different data generating processes, $(X, Y_{H_{t-1}}) \sim D_{t-1}$ and $(X, Y_{H_t}) \sim D_t$, respectively.

Without access to $Y^*$, the only feasible learning objective for a firm that wants to update its model parameters at time $t$ is the comparison between the latest human-ML collaborative labels with the new predictions.[1] For a given loss function $l : \mathcal{Y} \times \mathcal{Y} \to \mathbb{R}_+$ we can write this as follows:

$$L(Y_{M_t}, Y_{H_{t-1}}) := \underset{H \in \mathcal{H}}{\mathbb{E}} \big[ \underset{(X, Y_{H_{t-1}}) \sim D_{t-1}}{\mathbb{E}} l(Y_{M_t}, Y_{H_{t-1}}) \big]. \tag{1}$$

The firm wants to minimize the difference between the model predictions at time $t$ and the human labels at time $t - 1$. We can write the firm's problem as selecting a model $M_t$ to minimize the loss function in Equation 1:

$$\underset{M_t \in \mathcal{M}}{minimize} \, L(Y_{M_t}, Y_{H_{t-1}}). \tag{2}$$

For simplicity, we assume that at each time $t$, the firm collects enough data to perfectly learn the human-ML solution. In other words, with the optimal model, $L(Y_{M_t}, Y_{H_{t-1}}) = 0$. We discuss relaxing this assumption in Appendix A.7.

---

[1] We assume that models at time $t$ are trained exclusively on data from the previous period $t - 1$, although we can generalize our setting to include any data points from 0 to $t - 1$.

**Utility.**     In our scenario, the firm cannot quantify the true quality of a solution $Y$ with respect to $Y^*$. The loss in Equation 2 is just a surrogate for the loss $L(Y, Y^*)$, which is impossible or too costly to obtain. The firm thus defines the human label as "ground truth," and maximizes the similarity between model and human solutions, without knowing how close the human or ML solutions are to the indisputable ground truth. In order to evaluate the firm's progress in approximating $Y^*$, it is useful to define a measure of utility.

**Definition 1.** *(Utility) Let $d_X$ be a distance measure on $\mathcal{Y}$ with respect to a given $X \in \mathcal{X}$. The function $\mathbb{U} : \mathcal{X} \times \mathcal{Y} \to \mathbb{R}$ is a utility function on $\mathcal{X} \times \mathcal{Y}$, if $\forall X \in \mathcal{X}, Y_{min}, Y, Y', Y^* \in \mathcal{Y}$*

    1. $\exists Y_{min} \in \mathcal{Y} : \mathbb{U}(X, Y) \in [\mathbb{U}(X, Y_{min}), \mathbb{U}(X, Y^*)]$ *(bounded)*

    2. $\exists \varepsilon > 0 : |d_X(Y, Y^*) - d_X(Y', Y^*)| < \varepsilon \Rightarrow \mathbb{U}(X, Y) = \mathbb{U}(X, Y')$ *($\varepsilon$-sensitive)*

    3. $d_X(Y, Y^*) + \varepsilon < d_X(Y', Y^*) \Rightarrow \mathbb{U}(X, Y) > \mathbb{U}(X, Y')$ *(proximity measure)*

The utility of a solution for the firm is maximal if $Y$ is $\varepsilon$-close to $Y^*$ with respect to the underlying problem $X$. The variable $\varepsilon$ should be interpreted as the threshold below which a firm perceives no difference between two outcomes, i.e., it does not care about infinitely small improvements.

**Collaborative Characteristic Function.**     As time $t$ increases, the firm hopes that the distributions $D_t$ shift closer to the optimal distribution $D^*$, where $(X, Y) = (X, Y^*)$. In other words, for each model's distance $d$, $d(D_t, D^*) > d(D_{t+1}, D^*)$. This could happen, for example, if humans were able to easily compare available solutions and pick the one that is closest to the indisputable ground truth.

We can translate this continuous improvement into properties of the human decision function $H$ as follows: for all $t = 1, ..., T$ and $X \in \mathcal{X}$,

$$\mathop{\mathbb{E}}_{H \in \mathcal{H}}[\mathbb{U}(X, H(X, Y_{M_t}))] = \mathbb{U}(X, Y_{M_t}) + \delta_{M_t}. \tag{3}$$

The firm's hope is that $\delta_{M_t} \geq 0$ for $M_t$. Effectively, $\delta_{M_t}$ *characterizes* the human-ML collaboration for all utility levels of a model. If $\delta_{M_t}$ is positive, humans are able to improve on a ML prediction (and future model iterations will thus get better at approximating the ground truth). Instead, if $\delta_{M_t}$ is negative, humans will perform worse than the ML recommendations, and future model iterations will get progressively farther away from the ground truth.

We define the function given by Equation 3 as the collaborative characteristic function:

**Definition 2.** *(Collaborative Characteristic Function) For a utility function $\mathbb{U}$, humans $H \in \mathcal{H}$ and model $M \in \mathcal{M}$, we define the collaborative characteristic function $\Delta_{\mathbb{U}} : \mathbb{R} \times \mathcal{M} \to \mathbb{R}$ as follows:*

$$\Delta_{\mathbb{U}}(\mathbb{U}(X, Y_M), M) = \mathop{\mathbb{E}}_{H \in \mathcal{H}}[\mathbb{U}(X, H(X, Y_M))] = \mathbb{U}(X, Y_M) + \delta_M.$$

The function $\Delta_{\mathbb{U}}$ can take any arbitrary form. Several factors can affect $\Delta_{\mathbb{U}}$, e.g., ML explanations and monetary incentives (as we empirically explore in Section 4). Note that $\Delta_{\mathbb{U}}$ is also a function of the ML model $M$, and not just of its utility, because equal levels of utility across different ML models do not guarantee equal collaborative reactions from humans. In the rest of the paper however, we will shorten the notation and write $\Delta_{\mathbb{U}}(\mathbb{U}(X, Y_M))$.

**Collaborative Learning Path and Stable Points.**     Although $\Delta_{\mathbb{U}}$ has infinite support, a firm will only experience a discrete set of utility values achieved by humans with the help of ML recommendations. We call this the collaborative learning path. It is characterized by $\Delta_{\mathbb{U}}$, the utility of the first deployed model $s$, and the number of deployment cycles $T$:

**Definition 3.** *(Collaborative Learning Path) Let $\Delta_{\mathbb{U}}$ be a collaborative characteristic function, $t = 1, ..., T \in \mathbb{N}_{\geq 1}$ the number of deployment cycles and $s = \mathbb{U}(X, Y_{M_1})$ the utility of the starting model. We define the collaborative learning path to be the function*

$$\mathbb{L}_{\Delta_{\mathbb{U}}}(s, t) = \mathop{\mathbb{E}}_{H \in \mathcal{H}}\big[\mathop{\mathbb{E}}_{X \in \mathcal{X}}(\mathbb{U}(H(X, Y_{M_t})))\big].$$

**Definition 4.** *(Stable Point) A stable point $\mathbb{L}_{\Delta_{\mathbb{U}}}(s, t)$ occurs at $t$ if for all $t' \geq t$, $\mathbb{L}_{\Delta_{\mathbb{U}}}(s, t') = \mathbb{L}_{\Delta_{\mathbb{U}}}(s, t)$.*

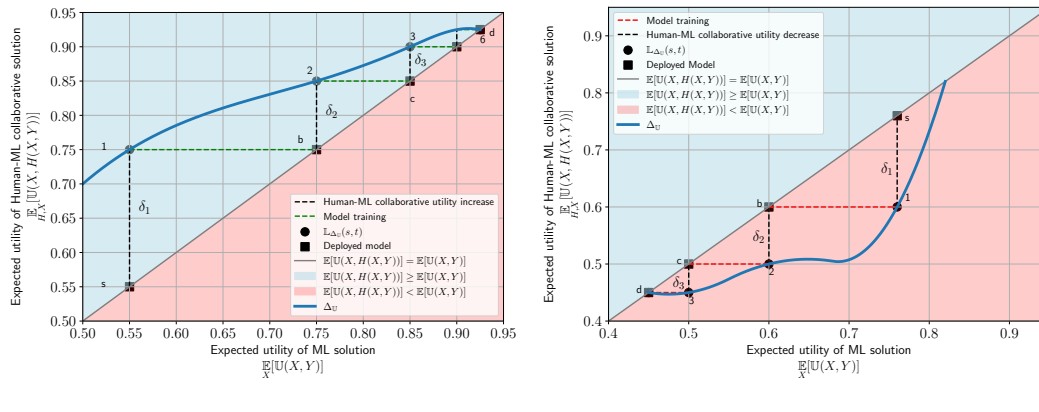

(a) Collaborative Improvement  (b) Collaborative Harm

Figure 1: **Collaborative Improvement (left)**: The firm's collaborative characteristic function and one collaborative learning path, if humans improve on the ML solution. The x-axis denotes the model expected utility, the y-axis denotes expected human+ML utility. The firm deploys a first model with utility (**s**). Then humans use the model and improve utility by $\delta_1$, leading to expected human+ML utility (**1**). The firm learns a new model with utility (**b**) on the new data distribution. This is viable under the assumption that the new model has the same utility as the previous period's human+ML labels, i.e., we can move horizontally from (**1**) to the 45-degree line at (**b**). Humans can further improve utility by $\delta_2$, which leads to expected utility (**2**). The dynamic improvement process continues until it reaches stable point utility (**6-d**). **Collaborative Harm (right)**: The firm deploys a model with expected utility (**s**) but the humans, when interacting with the model, decrease utility by $\delta_1$, with expected utility (**1**). The firm will thus learn a model of utility (**b**) on the new distribution. The downward spiral continues until stable point (**d**).

Stable points are states where the utility remains constant in all future model deployments. If $Y^*$ is unique for all $X$, then this is also a stable point for the distribution shifts. Whether a firm can reach a stable point on its collaborative learning function depends on the shape of $\Delta_{\mathbb{U}}$ and the initial model utility $s$. Figure 1 shows two examples of collaborative characteristic functions and collaborative learning paths. The 45-degree line includes the points where $\underset{X,H}{\mathbb{E}}[\mathbb{U}(X, H(X,Y))] = \underset{X}{\mathbb{E}}[\mathbb{U}(X,Y)]$

and maps the human performance at $t-1$ to the ML performance at $t$ under the assumption of perfect learning (i.e., $L(Y_{M_t}, Y_{H_{t-1}}) = 0$). Stable points will always lie on this line, because a stable point requires $\delta_t \approx 0$ ($|\delta_t| \leq \epsilon$), where $\epsilon$ is defined in Appendix A.5 and denotes the smallest change in utility that is possible for a given $\varepsilon$ from Definition 1. If $|\delta_t| > \epsilon$, it indicates that humans' influence changes labels $Y$ relative to the most recent ML model, leading to a new data distribution. The model at $t+1$ will thus differ from $M_t$, preventing stability. When the model and human+ML labels differ, there are two possible cases. First, $\delta_{M_t} > \epsilon$, which implies that the collaborative characteristic function $\Delta_{\mathbb{U}}$ is above the 45-degree line on that portion of the domain (Figure 1a). In this case, human+ML labels are closer to the indisputable ground truth than the model alone, which leads to improvements of subsequent model deployments. Second, if $\delta_{M_t} < -\epsilon$, the collaborative characteristic function is below the 45-degree line (Figure 1b). In this case, human+ML labels are further away from the indisputable ground truth than the model alone, which leads to deterioration of subsequent model deployments. We present the best-case and worst-case scenarios from Figure 1 as Propositions 1 and 2 below:

**Proposition 1.** *(Collaborative Improvement) If $\Delta_{\mathbb{U}}(\mathbb{U}(X, Y_M)) \geq \mathbb{U}(X, Y_M)$ for all $M \in \mathcal{M}, X \in \mathcal{X}$. Then $\mathbb{L}_{\Delta_{\mathbb{U}}}(s,t)$, is non-decreasing with $t = 1, ..., T$ and for sufficiently large $T$ it exists a $t' \in [1, T]$ such that $\mathbb{L}_{\Delta_{\mathbb{U}}}(s, t')$ is a stable point.*

*Proof.* (sketch) Because $\mathbb{U}$ is bounded, $\delta_M$ must be 0 in the extreme points. Furthermore, because of the $\varepsilon$-sensitivity of $\mathbb{U}$, the steps $t$ until reaching the maximum utility are also bounded. It follows that there exists a $t \in \mathbb{N}$ such that $\mathbb{L}_{\Delta_{\mathbb{U}}}(s,t) - \mathbb{L}_{\Delta_{\mathbb{U}}}(s, t+1) = 0$, which is a stable point. See Appendix A.6 for the complete proof. □

**Proposition 2.** *(Collaborative Harm) If $\Delta_{\mathbb{U}}(\mathbb{U}(X, Y_M)) \leq \mathbb{U}(X, Y_M)$ for all $M \in \mathcal{M}, X \in \mathcal{X}$. Then $\mathbb{L}_{\Delta_{\mathbb{U}}}(s,t)$, is non-increasing with $t = 1, ..., T$ and for sufficiently large $T$ it exists a $t' \in [1, T]$ such that $\mathbb{L}_{\Delta_{\mathbb{U}}}(s, t')$ is a stable point.*

*Proof.* Similar to the proof of Proposition 1. □

In practice, a firm's collaborative characteristic function can take any arbitrary shape, with portions above and portions below the 45-degree line. As long as the function is continuous, at least one stable point exists, and possibly more. When more than one stable point exist, the firm would like to reach the stable point with the highest utility (i.e., the highest point of the characteristic function lying on the 45-degree line). However, since the firm does not have access to the indisputable ground truth, when it reaches a stable point it does not know where such point lies on the 45-degree line.

In what follows, we offer a proof of concept of our theoretical setup. We empirically explore a context where it is easy for us to identify the indisputable ground truth. Although with some simplifying assumptions, the setting allows us to approximate a portion of the collaborative characteristic function, and explore the effects of human behavior on its shape, particularly the effect of monetary incentives and alternative solution selection criteria. We present study participants with instances of hard knapsack problems to answer the following research questions:

**RQ1:** *How do monetary incentives affect human performance?* To keep our treatment condition manageable, we explore the effect of different levels of performance bonuses on $\mathbb{U}(H(X, .))$, i.e., the human performance without ML recommendations.

**RQ2:** *Can we approximate the human-ML collaborative characteristic function $\Delta_{\mathbb{U}}$?* Here, we hold the performance bonus constant, and test humans' effect $\delta_M$ on utility for different levels of ML performance. This will enable us to construct two approximations of $\Delta_{\mathbb{U}}$ for a specific task.

## 4 EXPERIMENTAL SETUP

In this section, we describe our user study. The goal of our experimental setup is to simulate an environment in which users work on difficult tasks with the help of ML. The company responsible for deploying ML models does not know the optimal solution $Y^*$ (e.g., the true patient's diagnosis), and it trains ML models to replicate experts' decisions (doctor diagnoses). To evaluate how the company's models perform against $Y^*$, we need a setting in which we, as researchers, know the quality of any solution Y using a utility function $\mathbb{U}(X, Y)$. This allows us to make absolute quality assessments of solutions. Note that this is often unattainable in practice, as we argued in the introduction. The knapsack problem is particularly well suited for this context.

**The Knapsack Problem.** In our experiment, users solve instances of the knapsack problem. An instance involves selecting which of $n = 18$ items to pack into a knapsack, each with a weight $w$ and a value $v$. The objective is to maximize value without exceeding the weight limit $W$ of the knapsack (between 5 and 250). We focus on the one-dimensional 0-1 knapsack problem, in which participants choose which items to pack (see Appendix A.2 for a formal definition). We constrain the weights, values, and capacity of our instances to integer values, to make them easier to interpret by humans. We describe the details of the knapsack problem generation in Appendix A.10.

The knapsack problem has desirable properties for the empirical application of our framework. First, users do not require special training—beyond a short tutorial—to find a solution to the problem. Yet, the task is hard for humans, especially with a growing number of items (Murawski & Bossaerts, 2016). Thus, the optimal solution $Y^*$ is not obvious. Second, we can generate solutions to the knapsack problem in two ways. The "optimal" solution can be found with dynamic programming. The "ML" solution can be found by imitating what humans select and computing the training loss as the difference between the items selected by participants versus items selected by a model. Finally, it allows us to showcase our theoretical approach with two different utility functions and two selection criteria to approximate the collaborative characteristic function.

This setup allows us to quantify the utility of the proposed solution relative to the optimal solution. We define utility for the knapsack problem as follows:

**Definition 5.** *(Economic Performance) For a knapsack instance $X = ((w_1, \cdots, w_n), (v_1, \cdots, v_n), W)$ with optimal solution $\max_{x_1, \cdot, x_n; \sum_{i=1}^{n} x_i w_i \leq W} \sum_{i=1}^{n} x_i v_i =: Y^*$ and a valid solution Y we call the function $\mathbb{U}_{Econ}(X, Y) = \frac{Y}{Y^*}$ the economic performance of Y given X.*

Appendix A.4 contains details about $\mathbb{U}_{\text{Econ}}(X, Y)$ and discusses our results using an alternative utility function $\mathbb{U}_{\text{Opt}}(X, Y)$ (optimality), which is equal to one if a solution is optimal and zero otherwise. Note that there can be multiple optimal combinations of items to pack, but the optimal value $Y^*$ is always unique.

| Model | None | q1 | q2 | q3 | q4 | q5 | q6 |
|---|---|---|---|---|---|---|---|
| **Mean** $\mathbb{U}_{\text{Econ}}(X, Y)$ | . | 0.717 | 0.800 | 0.844 | 0.884 | 0.899 | 0.920 |
| **SD** | . | 0.083 | 0.105 | 0.098 | 0.105 | 0.088 | 0.085 |
| **No Bonus** | N=102 | | | | | | |
| **2-cent Bonus** | N=98 | | | | | | |
| **10-cent Bonus**[*] | N=100+117 | N=64 | N=78 | N=194 | N=179 | N=70 | N=191 |
| **20-cent Bonus** | N=96 | | | | | | |

Table 1: Matrix of treatment conditions. The columns denote information on the ML recommendation performance. The rows denote bonus payments for performance. The number of study participants are presented in the relevant cells. [*]We ran the 10-cent bonus treatment with no ML recommendation twice: once without a comprehension quiz for the bonus structure (100 participants) and once with the comprehension quiz (117).

**Study Design.** We recruited participants from Prolific[2] exclusively from the UK to ensure familiarity with the currency and weight metrics used to describe the knapsack items and monetary incentives in the study. Appendix A.11 presents screenshots of the web interface for each step of the study. At the beginning of the study, participants received a tutorial on the knapsack problem, our web application's interface, and the payment structure, described below. After the tutorial, the participants solved two practice problems and received feedback on their submission's performance. For the main task, each participant received 10 knapsack problems generated by Algorithm 1. For each problem, they had 3 minutes to submit their solution. If the participant did not actively click on the submit button, the selected items were automatically submitted at the 3-minute mark. Participants could take unlimited breaks between problems. At the end of the study, we asked participants about their demographics, previous experience with the knapsack problem, and how much effort they put in solving the task.

A total of 1,408 participants completed the study; we removed 119 participants due to forbidden browser reloads or uses of the browser's back-button, which left 1,289 for the analyses below. See Appendix A.9 for an overview of participants' demographics. On average, participants' compensation implied an hourly wage of £12.17 ($15.22), which is above the UK minimum wage of £11.44. Appendix A.3 contains additional payment details.

Every participant received a base payment of £2.00 (approx. $2.50) if they achieved at least 70% of the value of the optimal solution, averaged across the 10 knapsack instances they solved. We set the 70% threshold to discourage participants from randomly selecting items, as randomly-generated solutions that pick items until reaching the weight capacity have an average $\mathbb{U}_{\text{Econ}}$ around 60%.

Participants were randomly allocated into four monetary treatments and seven algorithmic recommendations (see Table 1). All monetary conditions were tested while users had no access to algorithmic recommendations. Participants in the **No Bonus** condition did not receive any additional payments beyond the base payment. Participants in the **2-cent Bonus** condition received an additional £0.02 for each percentage point of $\mathbb{U}_{\text{Econ}}$ above 70%. For example, if a participant achieved on average $\mathbb{U}_{\text{Econ}} = 85\%$, they would receive £2.00 + 15 × £0.02 = £2.30. Participants in the **10-cent Bonus** and **20-cent Bonus** treatments had similar incentives for performance, but higher monetary rewards for each additional percentage point increase in performance (£0.10 and £0.20, respectively).

We ran the **10-cent Bonus** treatment twice. In the second round, we introduced a comprehension quiz to ensure that our participants understood the payment structure. Within the **10-cent Bonus** with bonus comprehension quiz, we randomized access to ML recommendations. Users were randomly allocated to one of seven ML treatments. The control group had no ML recommendations. The other six groups had access to recommendations from progressively better ML models, denoted **q1** through **q6** as Table 1 shows on each of the last six columns.

The rationale for selecting the treatment conditions described above is the following. First, we want to understand whether monetary incentives change human effort, which in turn would translate into a shift in the collaborative characteristic function from Figure 1. Although ideally one would want to approximate the entire collaborative characteristic function under different incentive structures, to ensure statistical power under a limited budget, we opted for testing the role of varying bonuses without ML recommendations. Second, we want to understand how ML models of varying performance affect the human-ML performance to draw the collaborative characteristic function. Ideally, one would want these models to be trained on human labels (themselves potentially affected by previous model iterations) to mirror the theoretical framework, but this would have required sequential rounds

---

[2]https://www.prolific.com/

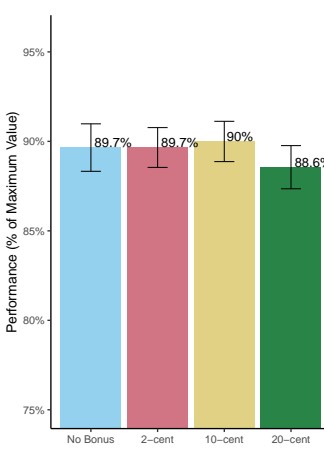

(a) Different bonus incentives.

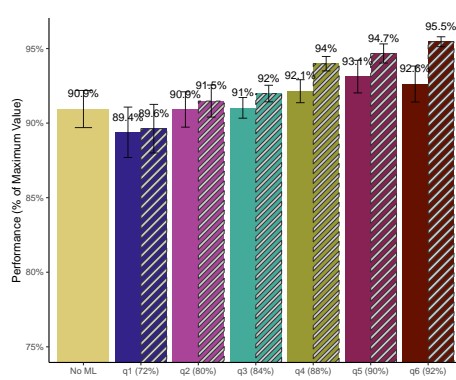

(b) Different ML recommendations.

Figure 2: Economic Performance Across Treatments. Error bars denote 95% confidence intervals based on standard errors clustered at the user level. Solid bars denote the average economic performance of the submitted solution, striped bars denote the performance if one picked the higher solution between the submitted solution and the provided ML recommendation. Appendix Figure 4 replicates the analysis using optimality as a measure of utility.

of experimentation. Instead, we approximate the sequential nature of our framework by training all models on optimally solved knapsack instances, rather than instances solved by humans. Appendix A.8 discusses further details on the model training. Ex-post, we verify that models trained on human labels (from the data collected during the experiment) have a wide range of utility levels. Appendix Figure 6 confirms that the utilities of the models selected for our treatments fall comfortably within the large range of utility levels of models trained on human labels. Nonetheless, we emphasize that these design choices limit our ability to truly replicate our theoretical model. We return to the limitations of this approach in the conclusion.

## 5 RESULTS

We start by discussing the null results of monetary performance incentives (**RQ1**). Figure 2a shows the results. On average, user economic performance without any bonus is 89.7% (light blue bar). None of the bonus alternatives are statistically distinguishable from the control group, nor from each other, and their point estimates are all between 88.6% (for the 20-cent bonus) and 90% (for the 10-cent bonus).

The null effect of monetary incentives is not due to the fact that users did not understand the bonus structure. To test this hypothesis, we can compare the performance of users in the two 10-cent bonus treatments without algorithmic recommendations (third column in Figure 2a and first column in Figure 2b, both yellow). These two treatments only differ by the fact that the one in Figure 2b had a comprehension quiz for the bonus structure. The difference in performance between the two treatments is a mere 0.9%, not statistically different from zero ($p = 0.268$, based on standard errors clustered at the user level). If we assume that the effect of monetary incentives without ML support is greater than or equal to their effect with ML support, these results imply that monetary incentives are unlikely to shift the collaborative characteristic function.

**RQ2:** We test the introduction of ML recommendations with a single bonus structure, the 10-cent bonus. Figure 2b presents the results. Focusing on the solid bars, three insights are noteworthy. First, comparing the first two columns (yellow and blue), models with low economic performance seem to lead humans to perform slightly worse than if they were not supported by ML recommendations (89.4% versus 90.9%). This comparison is not statistically significant ($p = 0.147$), likely due to low statistical power, but the level difference is not trivial (especially when looking at optimality as a measure of utility in Appendix Figure 4). Despite this, humans' utility does improve relative to the algorithmic recommendations (89.4% versus 71.8% in the q1 treatment, $p = 1.8e\text{-}28$). Second, models with better economic performance lead to increases in human performance, as evidenced by the progressively increasing economic performance from q1 to q6. Third, even if human performance

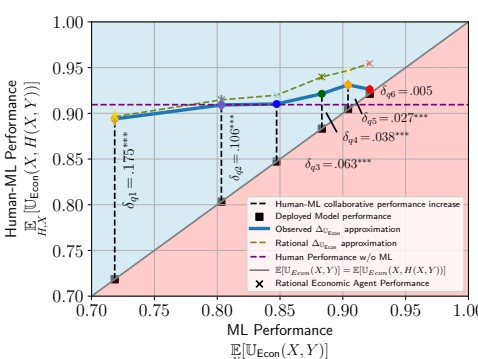 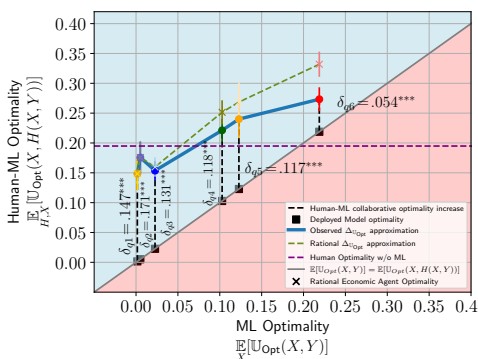

Figure 3: Empirical approximations of collaborative characteristic functions for two utility functions: economic performance (left) and optimality (right). Error bars represent confidence intervals based on participant-level clustered standard errors. Significance levels for the estimates of $\delta_{qi}$ are based on t-tests against the null $\delta_{qi} = 0$. ***: $p < 0.001$

increases with the performance of the ML recommendation, the increments in performance are quantitatively fairly small and sometimes statistically indistinguishable from one another, going from 89.4% when the model's performance is 72%, to 92.6% when the model's performance is 92%.[3] To evaluate whether the results are at least in part due to users changing their effort level, Appendix Figure 21 plots time spent on each problem across treatment conditions, and shows no clear patterns. Additionally, Appendix Figure 10 shows similarly high reported effort levels across ML and non-ML conditions.[4]

## 5.1 Approximation of the collaborative characteristic function

Figure 3 embeds our empirical results in the framework presented in Section 3. On the x-axis, we plot the economic performance of the six ML models deployed in our study. On the y-axis, we plot the performance of the solutions submitted by humans who receive ML recommendations: economic performance on the left plot and optimality on the right plot. Each of the points correspond to the six ML treatments of Figure 2b. We linearly interpolate the estimated points to form an approximation of the collaborative characteristic function $\Delta_{\mathbb{U}}$ (solid blue line). Looking at the left plot, in this approximation of a collaborative characteristic function, humans improve on the ML recommendations for ML performance levels between 70% and 92%. The estimated $\delta_{qi}$'s range from 17.5% ($p = 1.8e\text{-}28$) for $q1$, to 0.5% ($p = 0.46$) for $q6$. We denote $q6$ a stable point since the human improvement is estimated to be small and statistically indistinguishable from zero. The results imply that, for this portion of the domain, a firm could deploy a model with below-human performance and still converge to a stable point with 92% performance in subsequent deployments. The insights from the right plot are qualitatively similar, although there is no stable point in the portion of the domain that we explored.

An adjustment to the solution selection method allows us to simulate an additional collaborative characteristic function. Indeed, in this specific setting, as participants add items to the knapsack, in principle, they can easily compare the value of their solution to the value of the ML recommendation (both of which appear at the top of the interface, see Appendix Figure 18). If humans had picked the highest between their solution and the ML recommendation, the collaborative characteristic function would have shifted upward to the dashed green line in Figure 3, and the stable point would have achieved even higher performance. The discrepancy between the solid and dashed lines increases as the ML model improves, suggesting that even in a straightforward comparison, humans do not follow ML recommendations when it is in their best financial interest to do so (the difference can also be seen by comparing the solid and striped bars in Figure 2b). Appendix Figure 22 decomposes

---

[3]Regression results, controlling for time taken to solve each problem, are presented in Appendix Table 3.

[4]Appendix Figure 10 highlights an interesting contrast between users with and without ML recommendations. Indeed, participants without ML stated that they would have exerted less effort if they had been given ML recommendations. In contrast, the majority of those who received ML recommendations believed they would have exerted similar effort even without ML.

the net effect into two parts. On one hand, as the model performance improves, humans are more likely to follow its recommendations. On the other, when they do not follow the ML recommendation, as the model performance improves, it is much more likely that the submitted solution is inferior compared to the recommendation. Under both solid and dashed collaborative characteristic functions, we can imagine possible collaborative learning paths, $\mathbb{L}_{\Delta_\mathbb{U}}$. With this shape of $\Delta_\mathbb{U}$, the deployment decision is simple: all collaborative learning paths will eventually reach a stable point at above human performance.

## 6 CONCLUSIONS

We present a theoretical framework for human-ML collaboration in a dynamic setting where human labels can deviate from the indisputable ground truth. We introduce the collaborative characteristic function, which theoretically links the utility of ML models with respect to the indisputable ground truth, to the utility of humans using those same ML models to support their decisions. The collaborative characteristic function allows for multiple collaborative learning paths, depending on the utility of the initially deployed ML model. Each of the collaborative learning paths characterizes a possible ML deployment strategy and its ensuing dynamic learning process. We theoretically show conditions under which this dynamic system reaches a stable point through dynamic utility improvement or deterioration. We then present the empirical results of a large user study, which allows us to approximate collaborative characteristic functions of the knapsack problem. For ML models of economic performance between 72% and 92%, our empirical approximations of collaborative characteristic functions all lie above the 45-degree line. Any collaborative learning path starting at utility between 72% and 92% will thus likely converge to a stable point with utility around 92%. We explore two factors that can shift the collaborative characteristic function. We find that monetary incentives do not seem to affect human performance. However, we find that wherever applicable, a simple post-processing step that picks the best among available solutions (as is possible for the knapsack problem) can substantially shift the collaborative characteristic function upward, leading to stable equilibria of higher utility.

Our work has a number of limitations. On the theoretical side, our collaborative learning paths assume that the firm is able to perfectly replicate human+ML performance in future ML models. Appendix A.7 discusses stability when learning does not exactly replicate previous human+ML performance. However, since this assumption will likely not hold in the real world, imperfect learning may require more iterations than perfect learning, so more empirical studies are required to explore the speed of model convergence. On the empirical side, to reduce costs while maintaining statistical power, we only randomized monetary incentives without ML recommendations, and we randomized the quality of ML recommendations while fixing monetary incentives. Studying the interaction of monetary incentives and ML performance is an important extension. The null result of monetary incentives should be interpreted within our context. Specifically, the study participants received payments above minimum wage, and we only tested different levels of linear performance bonuses. It would be valuable to extend our work to evaluate the extent to which alternative base payments or non-linear bonuses may induce different levels of quality and effort by participants and thus collaborative characteristic functions of varying shapes.

Our approximation of $\Delta_\mathbb{U}$ for the knapsack problem is naturally incomplete for two main reasons. First, we use prediction models trained on synthetic data to approximate the collaborative characteristic function. Second, we did not test every possible level of model performance to fully draw the collaborative characteristic function. It is unlikely that these models and their linear interpolation would lead to the same performative trajectories as models trained on human feedback. We see this as a first proof of concept of collaborative characteristic functions, but much more work is needed to estimate these functions in real-world settings.

Future work could investigate the properties of $\Delta_\mathbb{U}$ that guarantee a unique optimal stable point, both theoretically and empirically. Provided that researchers have access to the indisputable ground truth, realistic empirical investigations of collaborative characteristic functions are crucial to shed light on the shape of those functions for practically relevant tasks such as medical diagnoses or hiring decisions. Future work should also discuss fairness aspects of this framework, e.g., whether or not fair stable points exist and how a firm can reach them. More generally, we hope this work generates more interest in studying settings where ML deployments lead to changes in the data generating process, which have broad managerial and practical applications.

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

# A APPENDIX

## A.1 DATA AND CODE

The code for model training, data generation, the web application for user study and our data analysis and plotting can be found in **retracted for anonymity; all files are part of the submission as .zip file**

## A.2 THE KNAPSACK PROBLEM

**Definition 6.** *(0-1 knapsack Problem) We call* $maximize \sum_{i=1}^{n} v_i x_i$ *s.t.* $\sum_{i=1}^{n} w_i x_i \leq W$ *with* $x_i \in \{0,1\}, v_i, w_i, W \in \mathbb{R}_+$ *the 0-1 knapsack Problem.*

## A.3 PAYMENT DETAILS

We calculated the base payment assuming an average time of 19 minutes to complete the study. The base payment was adjusted upward if the median time to completion was longer than 19 minutes. We adjusted the payment despite the fact that many participants finished our survey but did not enter the completion code directly afterwards. This sometimes increased the median time to completion.

## A.4 ANALYSIS WITH OPTIMALITY

**Definition 7.** *(Optimality) For a knapsack instance* $X$ *with optimal solution* $Y^*$ *and a valid solution* $Y$ *we call the function* $\mathbb{U}_{Opt}(X,Y) = \begin{cases} 1 & \text{if, } Y = Y^*, \\ 0 & \text{else} \end{cases}$, *the optimality of* $Y$ *given* $X$. *Furthermore, we call* $\mathbb{E}_{X}[\mathbb{U}_{Opt}(X,Y)]$, *the optimal solution rate over all* $X$.

**Observation 1.** *Economic performance and Optimality are utility functions (1).*

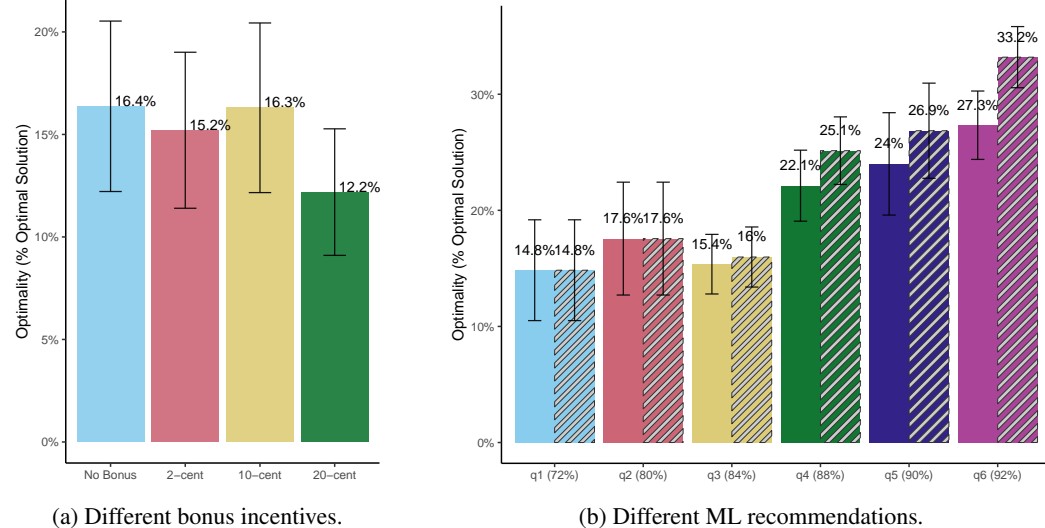

(a) Different bonus incentives.    (b) Different ML recommendations.

Figure 4: Optimality Across Treatments. Error bars denote 95% confidence intervals based on standard errors clustered at the user level. Solid bars denote the average optimality of the submitted solution, striped bars denote the optimality if one picked the best solution between the submitted solution and the provided ML recommendation.

*Proof.* We start with the proof that Economic Performance is a utility function. 1) Economic performance is bounded between 0 (for an empty knapsack) and 1, for the optimal value of the knapsack. 2) There exists an $\varepsilon > 0$, which is the minimum value of an item for the knapsack problem. The value of that item is the smallest possible distance between two solutions which are not equally good. 3) Because the $Y$ in our case is the sum of the values of the items in the knapsack and $Y*$ is the maximum possible value of the knapsack, any value that is closer to the optimal solution has also higher economic performance because the numerator grows. We chose $\varepsilon$ to be the minimum item value, thus this minimum increase in value between solutions is fulfilled. In summary, Economic Performance satisfies all three criteria of a utiltiy function.

We continue with the proof that optimality is a utility function. 1) it is 0 or 1 and thus bounded. 2) If we choose $0 < \varepsilon < 1$, then $\varepsilon$-sensitivity is satisfied. 3) Is always true for the choice of our $\varepsilon$. Assume for example $\varepsilon = 0.5$, then it is that $d(1,1) + 0.5 < d(0,1)$ and $\mathbb{U}(1) > \mathbb{U}(0)$. This statement is true for all $0 < \varepsilon < 1$ which is what we specified for $\varepsilon$. □

Optimality is the function that indicates whether a solution to a knapsack problem has the optimal value or not. Figure 5 shows the empirical collaborative characteristic function for optimality as utility function. The humans achieve approximately 20% optimalty without ML advice. The effect of human on human-ML performance is significant for all models ($p < 0.001$). Interestingly, the effect is large even beyond human performance. Furthermore, for models q1,q2,3 with extremely low utility (average optimality of almost 0%), human effects on the overall outcome is large and close to human performance. As in Figure 3, the utility gain of rationally acting humans would have been larger for most models. Our observations suggest that stable points of optimality would lie above human performance without ML adivce.

A.5   COMMENTS ON THE DEFINITION OF UTILITY

We want to denote that $\varepsilon$-sensitivity implies the following:

**Observation 2.** $\exists \epsilon, \varepsilon > 0 : |d_X(Y, Y^*) - d_X(Y', Y^*)| = \varepsilon \Rightarrow |\mathbb{U}(X, Y) - \mathbb{U}(X, Y')| = \epsilon$

This means that there is a minimum utility change that we call $\epsilon$.

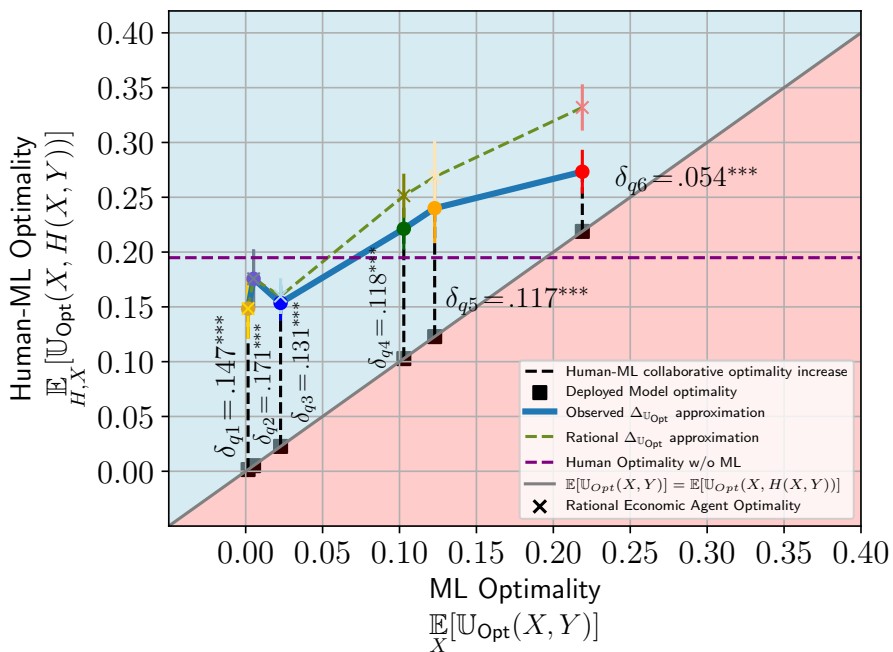

Figure 5: Empirical Collaborative Characteristic Function for the "Optimality" utility function. Confidence intervals are based on standard errors clustered at the participant level.

### A.6 PROOF OF PROPOSITON 1&2

**Proposition 1** (Collaborative Improvement)
*If $\Delta_{\mathbb{U}}(\mathbb{U}(X, Y_M)) \geq \mathbb{U}(X, Y_M)$ for all $M \in \mathcal{M}, X \in \mathcal{X}$. Then $\mathbb{L}_{\Delta_{\mathbb{U}}}(s, t)$, is non-decreasing with $t = 1, ..., T$ and for sufficiently large $T$ it exists a $t' \in [1, T]$ such that $\mathbb{L}_{\Delta_{\mathbb{U}}}(s, t')$ is a stable point.*

*Proof.* Let $t \in 1, ..., T$ be the number of deployment (epochs) that a firm will make. The firm perfectly learns the data distribution in every epoch, in other words, we assume that $L(Y_{M_t}, Y_{H_{t-1}} = 0, \forall t$. Furthermore, it is $\Delta_{\mathbb{U}}(\mathbb{U}(X, Y_M)) \geq \mathbb{U}(X, Y_M)$ for all $M \in \mathcal{M}, X \in \mathcal{X}$.

We first show that $\mathbb{L}_{\Delta_{\mathbb{U}}}(s, t)$ is non-decreasing with $t$. For that, assume that there exists $t$ for which $\mathbb{L}_{\Delta_{\mathbb{U}}}(s, t) > \mathbb{L}_{\Delta_{\mathbb{U}}}(s, t + 1)$. But $\mathbb{L}_{\Delta_{\mathbb{U}}}(s, t + 1) = \underset{X \in \mathcal{X}}{\mathbb{E}}(\mathbb{U}(H(X, Y_{M_{t+1}}))) \geq^{\delta_i \geq 0} \underset{X \in \mathcal{X}}{\mathbb{E}}(\mathbb{U}(Y_{M_{t+1}})) =^{L(Y_{M_{t+1}}, Y_{H_t})=0} \underset{X \in \mathcal{X}}{\mathbb{E}}(\mathbb{U}(H(X, Y_{M_t}))) = \mathbb{L}_{\Delta_{\mathbb{U}}}(s, t)$. It follows that $\mathbb{L}_{\Delta_{\mathbb{U}}}(s, t)$ must be non-decreasing.

Now we show that there exists a $t' \in [1, T]$ such that $\mathbb{L}_{\Delta_{\mathbb{U}}}(s, t')$ is a stable point for sufficiently large $T$. For this, consider that $\mathbb{U}$ has a maximum $\mathbb{U}(Y^*)$ (Property 1 (bounded) of definition 1) and there exists a minimum increment of utility $\epsilon$ (see A.5) in each deployment. If we do not achieve at least $\epsilon$ increment in utility, we have reached a stable point. Thus, we can write the maximum utility as $\mathbb{U}(Y^*) = \mathbb{U}(Y_{M_t}) + N\epsilon$. For sufficiently large ($T \geq N + 1$), this implies that we reached maximum utility with $\mathbb{L}_{\Delta_{\mathbb{U}}}(s, T)$, and every deployment beyond that must have equal utility. $\square$

**Proposition 2** (Collaborative Harm)
*If $\Delta_{\mathbb{U}}(\mathbb{U}(X, Y_M)) \leq \mathbb{U}(X, Y_M)$ for all $M \in \mathcal{M}, X \in \mathcal{X}$. Then $\mathbb{L}_{\Delta_{\mathbb{U}}}(s, t)$, is non-increasing with $t = 1, ..., T$ and for sufficiently large $T$ it exists a $t' \in [1, T]$ such that $\mathbb{L}_{\Delta_{\mathbb{U}}}(s, t')$ is a stable point.*

*Proof.* Analogous to the proof of Proposition 1. $\square$

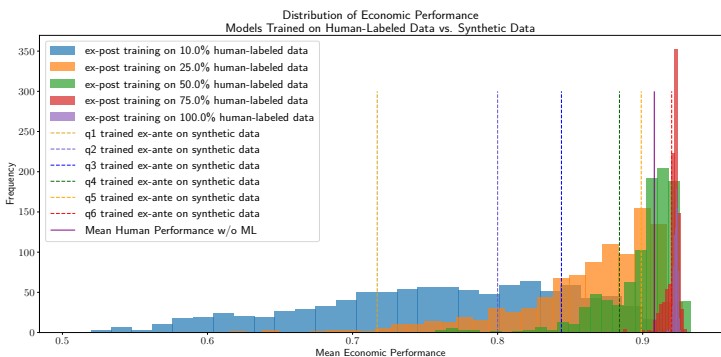

Figure 6: Distribution of mean model performances trained on human data ex-post to verify that we picked models with reasonable performances. Vertical lines indicate the economic performances of models trained on synthetic data, which were chosen to approximate the collaborative characteristic function of the task.

### A.7 PERFECT VS IMPERFECT LEARNING

In this section, we discuss what changes if we loosen the assumption $L(Y_{M_t}, Y_{H_{t-1}}) = 0$. We call this assumption the "perfect learner" assumption because the firm perfectly learns the human labels from epoch $t-1$ with a model in epoch $t$. In the following, we consider an imperfect learner such that $L(Y_{M_t}, Y_{H_{t-1}}) = \sigma$.

Figure 1 helps illustrate the relaxation of the assumption. An imperfect learner effectively amounts to tilting the 45-degree line upward or downward. The tilt is upward if imperfect learning leads the ML model to have lower performance with respect to the indisputable ground truth compared to the human (i.e., the slope of the straight line is higher than 1). The tilt is downward if imperfect learning leads the ML model to have higher performance with respect to the ground truth (i.e., the slope of the straight line is lower than 1).

It is straightforward to extend Proposition 1 and Proposition 2 to the case of imperfect learning. In the case of collaborative improvement $(\Delta_{\mathbb{U}}(\mathbb{U}(X, Y_M)) \geq \mathbb{U}(X, Y_M))$, the human will improve on any model that the firm can deploy. However, if imperfect learning leads to $(\mathbb{U}(Y_{M_t}) - \mathbb{U}(Y_{M_{t-1}})) < 0$ then the performance gain from the human effort does not fully transfer to the ML model. If the above statement is true for all $M_t$, then the imperfection creates collaborative harm, which is the case covered in Proposition 2. However, this would still lead to a stable point. The alternative scenario where $L(Y_{M_t}, Y_{H_{t-1}}) = \sigma \Rightarrow (\mathbb{U}(Y_{M_t}) - \mathbb{U}(Y_{M_{t-1}})) > 0$ for all $M_t$, is still a scenario of collaborative improvement, which means that we will again reach a stable point.

In summary, imperfect and perfect learners are analogous. In both cases, the crucial question is how much humans improve the system's performance. For the case of an imperfect learner, an additional empirical question is how much of the human improvement transfers to the ML model.

### A.8 MODEL TRAINING

We release the code required for training our models, our model parameters and all predictions for the instances together with the instances that participants saw. Learning to solve the knapsack problem is a research area for itself, however for the small, one-dimensional case of our experiment, it is possible on consumer hardware. We only train models for knapsack instances with 18 items. As input features we concatenate weights $w_1, ..., w_{18}$, values $v_1, ..., v_{18}$, the weight constraint $W$, the sum of the weights and the sum of the values. Thus, our input dimension is 39. Our goal was to train models with a broad spectrum of economic performances, not to solve the knapsack problem perfectly. We added 5 fully connected layers, 4 of them with ReLU activation functions. We use torch.Sigmoid() for our outputs. The output dimension was 18 and the output values in each index can be interpreted as the likelihood that the item belongs to a solution or not. For more details on the architecture, see our code. In summary, all models had dimensions in order of layers: (39,90), (90,550), (550,90), (90,84), (84,18).

We want to highlight two important aspects of how we thought about the model training. First, did not want to use any prior knowledge that a firm in our setting could not have either. For example,

if we could have known the utility of a knapsack solution (economic performance or optimality) we could have just directly maximized it, or if we could know the optimal solution, we could have just used the distance to the optimal solution as our loss. Instead, we used the binary cross-entropy between the label and prediction as our loss. The label was a 18-dimensional 0-1 vector. If the i-th entry of this output vector is 1, it means that the i-th item is in the knapsack and otherwise not. Thus we simply minimized the differences between chosen items in our training data and those of our model. For us, this was a reasonable analogy for the application context of healthcare in which every "item" is a diagnosis or a symptom (e.g. an ICD10 code).

Because our financial budget was limited and we wanted to test multiple models, we trained all models on optimally solved knapsack instances. It would have also created a lot of overhead and space for errors if we would have collected the data of model q1 then trained q2 and rerun the user study. Training them all on generated labels made it possible to run more treatments at once. We still wanted to use ML models instead of solutions produced with dynamic programming, because we wanted to incorporate the distributional character of ML predictions (see Figure 11) and study the reaction to different quantiles of solution quality in greater detail in future work.

However, we had to include two pieces of prior knowledge in order to achieve better model performance (especially for q5 and q6). First, we sorted the items by density (value/weight). This is a big advantage in general, but only a small one for our knapsack instances because weights and values are strongly correlated. Second, we normalized weights and values in a pre-processing step. In our setting, both operations could not have been done by the firm (what is a normalized symptom)? However, with those minor modifications we were able to create a larger range of models without massive resources and still just immitate the "human" label without incorporating anything in the loss. In a post-processing step, we sorted the items by sigmoid outputs. We then added items to the knapsack until the weight constraint was reached. From that item selection, we calculated the actual knapsack values. For more details, please visit our github repository **To be added after acceptance**.

## A.9 OVERVIEW STATISTICS

Figure 7 shows the overview of the answer to the demographic questions in the end of our study. Most participants held an Undergraduate degree, were between 25 and 44 years old and have not heard about the knapsack problem before completing the study. $50.1\%$ of the participants identified as female $48.6\%$ as male and $0.8\%$ as non-binary or non-gender conforming. $96.8\%$ of the participants have not heard about the knapsack problem before this study. Figure 8 shows the perceived difficulty of the task for the participants, as well as the reported effort the participants put to complete the task. Most participants perceived the task as neutral to hard and put in large to very large effort (self-reportedly). Figure 9 shows how much effort people think they would have spent with or without the help of ML. It seems like participants who had no ML help think they would put less effort in the task. People who had the help of ML reported to put about as much effort as all participants reported to put in right now. Future work should investigate these perceptions in detail.

## A.10 GENERATING HARD KNAPSACK PROBLEMS

knapsack problems where the weights $w_i$ and values $v_i$ are strongly, yet imperfectly, correlated (Pisinger, 2005; Murawski & Bossaerts, 2016) tend to be hard to solve. We generate knapsack instances with strong correlations ($r \in [0.89, 1.00]$, mean $r = .9814$) using Algorithm 1, following the criteria for difficult problems outlined by Pisinger (2005). In our experiment, users solve knapsack instances with $n = 18$ items, $W_{min} = 5$, $W_{max} = 250$. We constrain the weights, values, and capacity of our instances to integer values, to make them easier to interpret by humans.

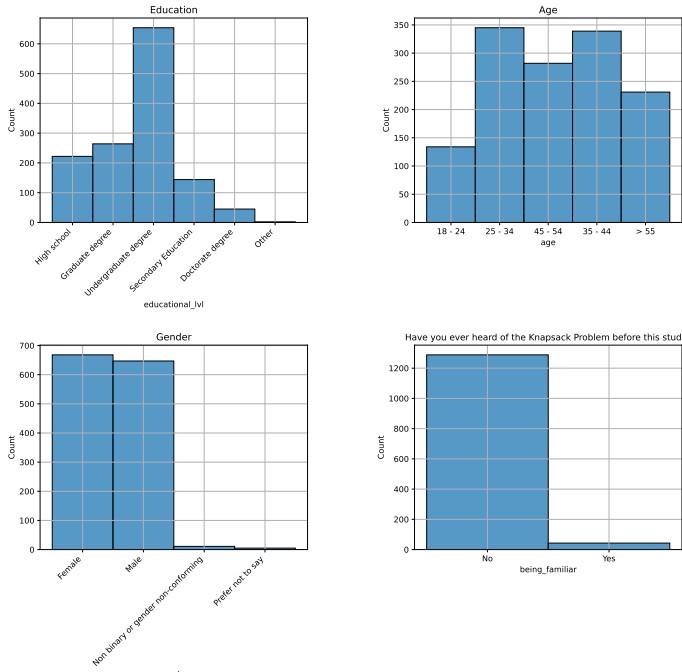

Figure 7: Highest level of education completed, age group, gender and whether participants have heard of the knapsack problem before this study.

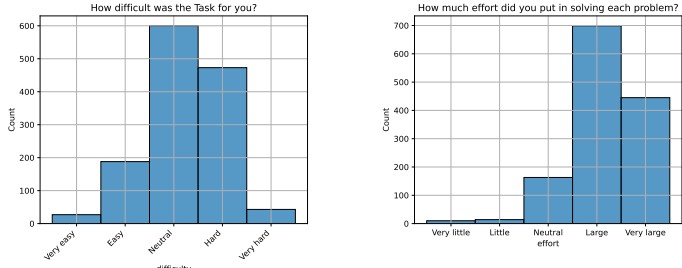

Figure 8: Perceived difficulty of the task versus the reported level of effort participants reported in our study

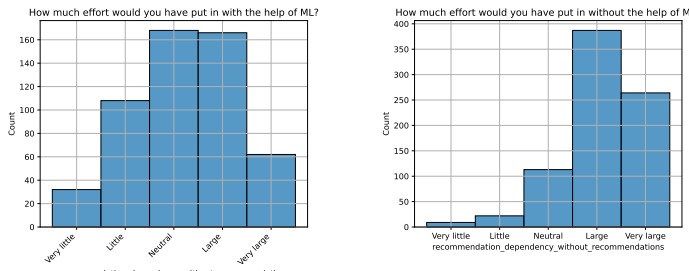

Figure 9: How much effort participants thought that they would have spent with/without the help of ML

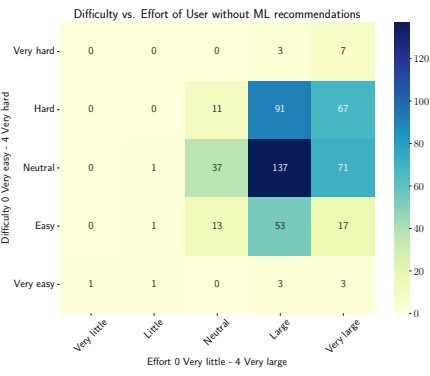

(a) Perceived difficulty vs effort without ML

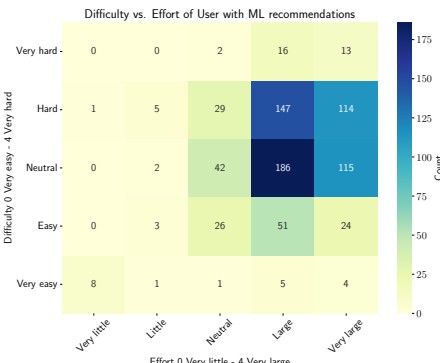

(b) Perceived difficulty vs effort with ML

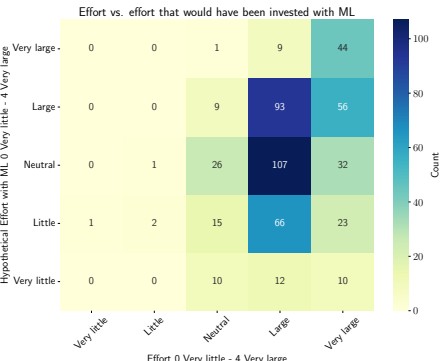

(c) Effort vs hypothetical effort with ML

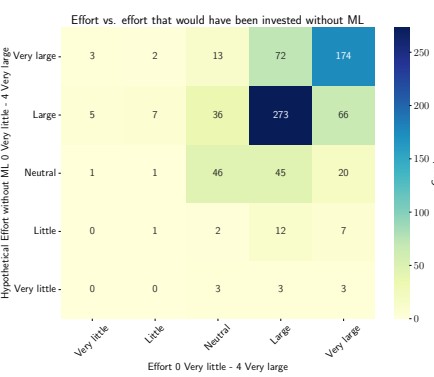

(d) Effort vs hypothetical effort without ML

Figure 10: Task Difficulty, Effort, and Hypothetical Effort with and without ML recommendations. Perceived difficulty of the task does not vary between participants with and without ML recommendations. Participants without ML recommendation expect to invest less effort if they would have had a ML recommendation, while participants with ML recommendations expected to invest the same effort without ML.

---

**Algorithm 1** Generate hard knapsack instance

---

**Require:** number of items $n \geq 0$, knapsack capacity range $W_{min}, W_{max} > 0$
$W \leftarrow random.uniform.integer(W_{min}, W_{max})$
$w \leftarrow random.uniform.integer(1, W, n)$ ▷ n-dimensional vector of weights
$i \leftarrow 1$
**while** $i \leq n$ **do**
$\quad v_i \leftarrow max(1, random.uniform.integer(w_i - \lfloor \frac{W}{10} \rfloor, w_i + \lfloor \frac{W}{10} \rfloor))$
**end while**

---

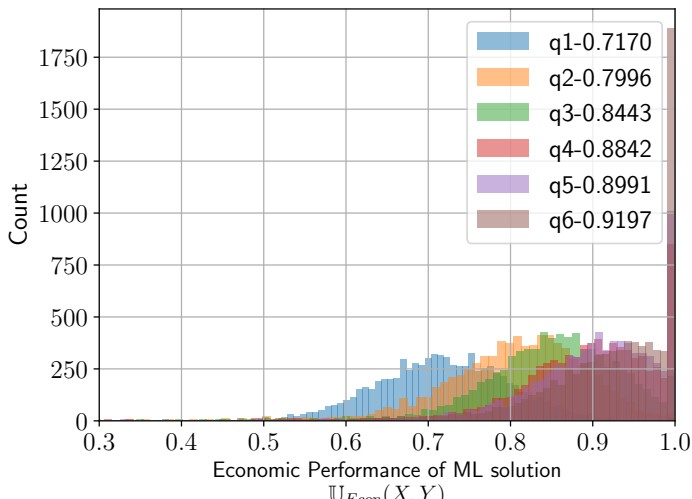

Figure 11: Distribution of economic performances of solutions by the six models we deployed in our experiment.

**Tutorial 1/5**

We will now explain the problem you have to tackle during this experiment. Please read the following pages carefully. In the end of this tutorial, you will receive two practice problems in order to get familiar with the interface.

**The Knapsack Problem**

- Imagine you have a backpack with a **maximum weight** it can hold.
- You also have several **items**, each with its own **weight and value**.
- The goal is to choose a combination of items to put in the backpack that **maximizes the total value**, without exceeding the backpack's maximum weight.
- It's like trying to pack your backpack with the most valuable things possible, making sure it does **not get too heavy** to carry.

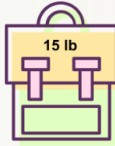

Figure 12: Tutorial 1/5

## A.11 SURVEY DESIGN

## A.12 FUTHER STATISTICAL INSIGHTS

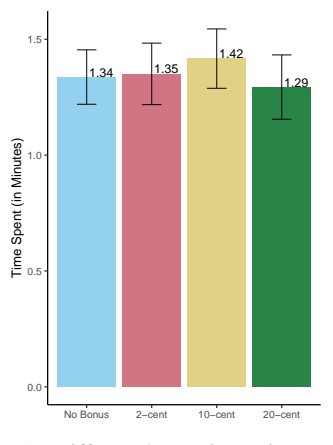

(a) Different bonus incentives.

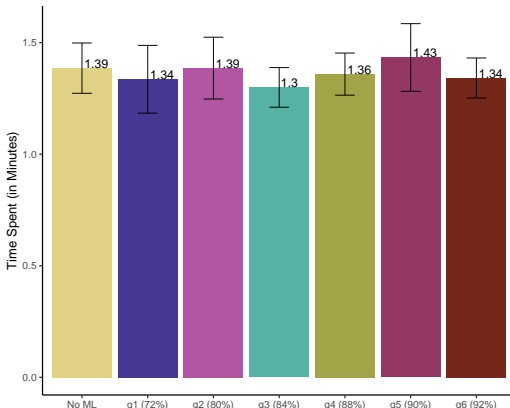

(b) Different ML recommendations.

Figure 21: Time Spent Across Treatment Conditions

**Tutorial 2/5**

**Let us solve an example!**

- In the picture below, you can see an example of a knapsack problem.
- The backpack has a **maximum weight of 15 lb** and we have four items to choose from.
- In this simple case, each item has a **weight of 5 lb.**
- Since all items weigh the same, it is quite easy to pick which items to put in the backpack to achieve the **maximum value**. We want to pick the three items that have the highest individual values. This means that we will pick the item on the left, on the bottom left and the item on the bottom right.

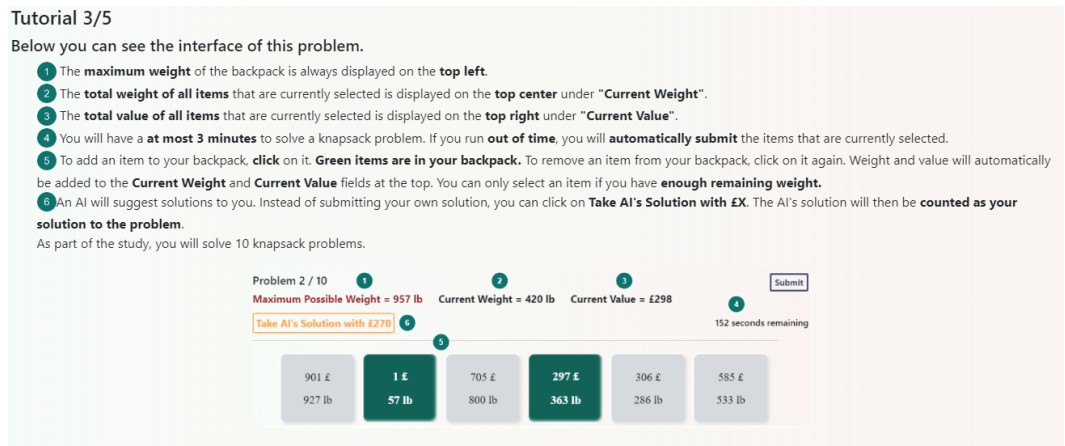

Figure 13: Tutorial 2/5

**Tutorial 3/5**

**Below you can see the interface of this problem.**

1. The **maximum weight** of the backpack is always displayed on the **top left**.
2. The **total weight of all items** that are currently selected is displayed on the **top center** under "**Current Weight**".
3. The **total value of all items** that are currently selected is displayed on the **top right** under "**Current Value**".
4. You will have a **at most 3 minutes** to solve a knapsack problem. If you run **out of time**, you will **automatically submit** the items that are currently selected.
5. To add an item to your backpack, **click** on it. **Green items are in your backpack.** To remove an item from your backpack, click on it again. Weight and value will automatically be added to the **Current Weight** and **Current Value** fields at the top. You can only select an item if you have **enough remaining weight.**
6. An AI will suggest solutions to you. Instead of submitting your own solution, you can click on **Take AI's Solution with £X**. The AI's solution will then be **counted as your solution to the problem**.

As part of the study, you will solve 10 knapsack problems.

Figure 14: Tutorial 3/5 (with ML treatment)

**Tutorial 4/5**

**Rewards and Payment:**

- You will receive a payment of **£2.00** if you complete all 10 knapsack problems and achieve at least 70% of the maximum achievable value on average over all 10 knapsack problems. This is a **low threshold** and you will have no problems reaching this value, if you take this study seriously.
- In the end, we will evaluate your average performance and **for each percentage point over 70%, you will receive a bonus payment of £0.10.**
- For example, if you **achieved on average 80% of the maximum achievable value**, you will get **£2.00 + 10*£0.10 = £3.00.** You can find some example payments in the table below.
- We will reveal your performance **after** you have solved all problems.

| Performance | Bonus | Total |
|---|---|---|
| 70% | £0,00 | £2,00 |
| 71% | £0,10 | £2,10 |
| 85% | £1,50 | £3,50 |
| 99% | £2,90 | £4,90 |
| 100% | £3,00 | £5,00 |

Figure 15: Tutorial 4/5 (with 10 cents/ppt monetary incentive)

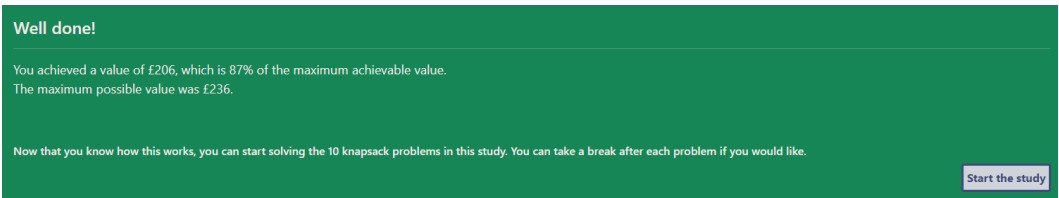

Figure 16: Tutorial 5/5 (with comprehension quiz)

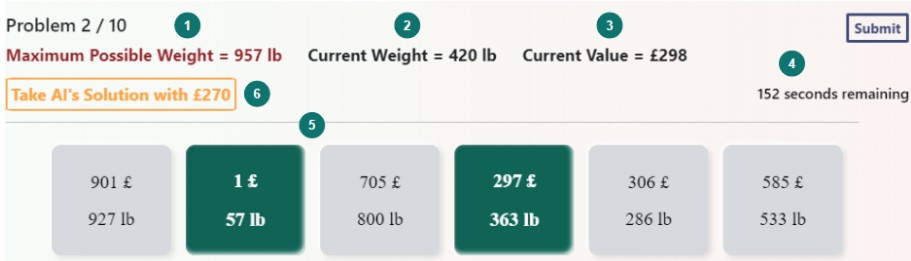

Figure 17: Feedback to a practice problem

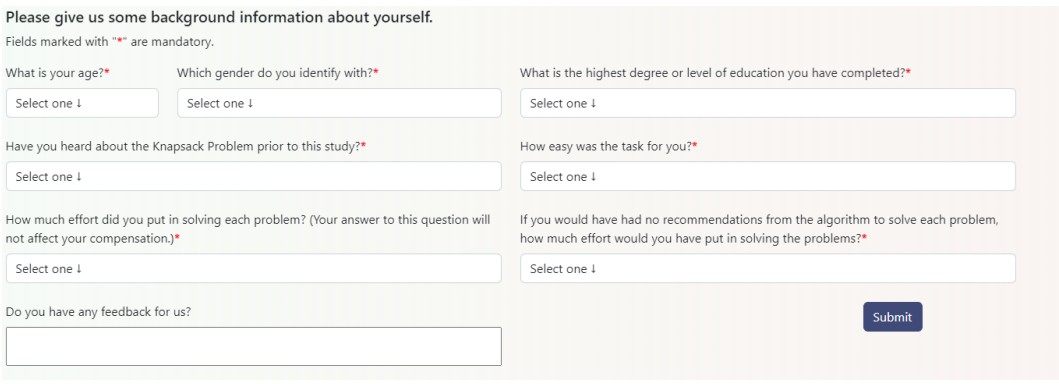

Figure 18: Interface for the main task: **1)** the knapsack capacity, **2)** sum of weights of selected items, **3)** sum of values of selected items, **4)** remaining time, **5)** items with weights and values, **6)** machine learning solution (only visible if user receives corresponding treatment). Clicking on gray items adds them to the knapsack if the weight allows it, and clicking on green items removes them from the knapsack. The total weight and value of selected items is shown at the top and automatically updated.

Figure 19: Demographic questions after tasks

Here you can see your performance on the knapsack problems:

| Game Number | Performance |
|---|---|
| 2 | 73 % |
| 3 | 74 % |
| 4 | 91 % |
| 5 | 81 % |
| 6 | 74 % |
| 7 | 74 % |
| 8 | 69 % |
| 9 | 57 % |
| 10 | 18 % |
| 11 | 61 % |

Average Performance: 67.20%

Figure 20: Score screen for performance feedback in the end

| | $\mathbb{U}_{\text{Econ}}(X, H(X))$ | $\mathbb{U}_{\text{Econ}}(X, H(X))$ | | $\mathbb{U}_{\text{Econ}}(X, H(X,Y))$ | $\mathbb{U}_{\text{Opt}}(X, H(X,Y))$ |
|---|---|---|---|---|---|
| Intercept | 0.7620*** | 0.6957*** | Intercept | 0.8082*** | −0.0297 |
| | (0.0184) | (0.0408) | | (0.0049) | (0.0245) |
| 02-cent bonus | 0.0003 | 0.1087* | q1 (72%) | −0.0131 | −0.0408 |
| | (0.0077) | (0.0484) | | (0.0044)** | (0.0260) |
| 10-Cent bonus | 0.0011 | 0.0683 | q2 (80%) | 0.0011 | −0.0161 |
| | (0.0076) | (0.0499) | | (0.0043) | (0.0283) |
| 20-Cent bonus | −0.0087 | 0.0787 | q3 (84%) | 0.0048 | −0.0324 |
| | (0.0079) | (0.0478) | | (0.0049) | (0.0198) |
| log(seconds spent) | 0.0322*** | 0.0481*** | q4 (88%) | 0.0151*** | 0.0333 |
| | (0.0039) | (0.0093) | | (0.0033) | (0.0214) |
| 02-cent bonus · log(seconds spent) | — | −0.0260* | q5 (90%) | 0.0247*** | 0.0518* |
| | | (0.0112) | | (0.0043) | (0.0263) |
| 10-cent bonus · log(seconds spent) | — | −0.0161 | q6 (92%) | 0.0211*** | 0.0880*** |
| | | (0.0115) | | (0.0033) | (0.0213) |
| 20-cent bonus · log(seconds spent) | — | −0.0210 | log(seconds spent) | 0.0240*** | 0.0533*** |
| | | (0.0113) | | (0.0010) | (0.0045) |
| $N$ | 3,960 | 3,960 | $N$ | 8,930 | 8,930 |
| Adj.$R^2$ | 0.0613 | 0.0661 | Adj.$R^2$ | 0.0733 | 0.0281 |

Table 2: Linear regressions with clustered standard errors on participant id. Effect of monetary incentive on $\mathbb{U}_{\text{Econ}}$ of human solutions **(left)**. Effect of ML recommendation on different levels of economic performance $\mathbb{U}_{\text{Econ}}$ **(right)**. Standard errors in parentheses. * $p < 0.05$, ** $p < 0.01$, *** $p < 0.001$.

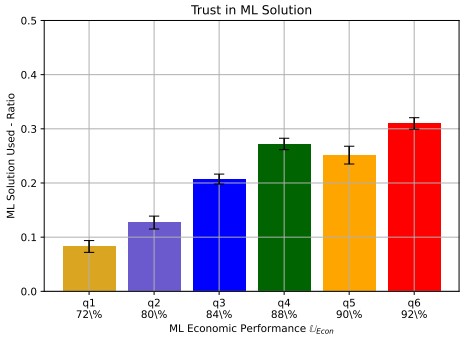

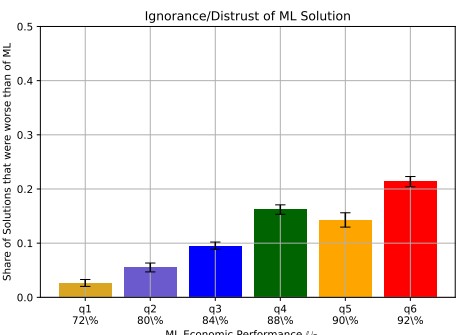

(a) Rate of ML advice usage increased with better performance.

(b) Participants ignore the ML recommendation with better performance.

Figure 22: ML-usage increased with better ML performance. Share of ignored ML solutions did also increase with better performance.

| | $\mathbb{U}_{\text{Econ}}(X, H(X))$ | | |
| --- | --- | --- | --- |
| | (1) | (2) | (3) |
| Intercept | 0.7620*** | 0.7676*** | 0.8082*** |
| | (0.0184) | (0.0276) | (0.0049) |
| 2-cent Bonus | 0.0003 | | |
| | (0.0077) | | |
| 10-cent Bonus | 0.0011 | | |
| | (0.0076) | | |
| 20-cent Bonus | −0.0087 | | |
| | (0.0079) | | |
| Comprehension Quiz | | 0.0104 | |
| | | (0.0076) | |
| q1 (72%) | | | −0.0131 |
| | | | (0.0044)** |
| q2 (80%) | | | 0.0011 |
| | | | (0.0043) |
| q3 (84%) | | | 0.0048 |
| | | | (0.0049) |
| q4 (88%) | | | 0.0151*** |
| | | | (0.0033) |
| q5 (90%) | | | 0.0247*** |
| | | | (0.0043) |
| q6 (92%) | | | 0.0211*** |
| | | | (0.0033) |
| log(seconds spent) | 0.0322*** | 0.0317*** | 0.0240*** |
| | (0.0039) | (0.0064) | (0.0010) |
| $N$ | 3,960 | 2170 | 8,930 |
| Adj.R$^2$ | 0.0613 | 0.0506 | 0.00733 |
| Included Bonus Treatments | All | 10-cent | 10-cent |
| Included ML Treatments | No ML | No ML | All ML |
| Comprehension Quiz | No | Both | Yes |

Table 3: Linear regressions of economic performance $\mathbb{U}_{\text{Econ}}$ on dummies for the various treatment conditions. Column 1 includes all treatment conditions without ML recommendations and without comprehension quiz. It tests the difference in performance across different bonus levels. Column 2 includes the two treatment conditions without ML recommendation and with 10-cent bonus. The difference between the two treatment conditions is the presence of a comprehension quiz for the bonus structure. Column 3 includes all treatments with comprehension quiz and 10-cent bonus. It tests the difference in performance across ML recommendations with different performance. Standard errors, in parentheses, are clustered at the participant level. * $p < 0.05$, ** $p < 0.01$, *** $p < 0.001$.

