# OpenReview forum: "A Dynamic Model of Performative Human-ML Collaboration: Theory and Empirical Evidence"
_ICLR.cc/2025/Conference — Submitted to ICLR 2025_

### Official Review · Reviewer_t43N · 2024-10-28

**Soundness:** 3
**Presentation:** 3
**Contribution:** 2
**Rating:** 5
**Confidence:** 3

**Summary:**

This paper presents a theoretical framework for describing the collaboration process between ML models and human decision-making. By defining a utility function and a collaborative characteristic function, it gives a sufficient condition for achieving a stable point in the optimal case. Additionally, an empirical experiment with real users offers interesting insights into practical applications.

**Strengths:**

1. The studied problem is important. Developing a collaborative system that integrates ML models with human decision-making to consistently achieve better outcomes is both a valuable and challenging topic for academia and industry.
2. A theoretical framework is proposed to describe the collaboration process and quantify the quality of solutions from both models and humans. A sufficient condition, which ensures non-decreasing utility, is provided to guarantee the achievement of a stable point.
3. An empirical experiment was conducted with real users, offering interesting insights into practical applications.

**Weaknesses:**

1.	The theoretical framework primarily aims to describe the problem. Both the theory and convergence conditions rely on acquiring the utility function, which seems to be merely achievable with the knowledge of the ground truth. However, as discussed in Introduction and Related Works, a key intuition of this paper is addressing the inaccessibility of ground truth in real-world scenarios. Consequently, the theory offers limited insights at the methodological level.
2.	Several expressions and derivations are unclear and lack rigor. E.g., it seems that $\delta_{M_t}$ in Eq. (3) should be determined jointly by ${M_t}$and $X$. In Definition 3 and Proof A.6, $U(H(X, Y_{M_t}))$ should be instead be written as  $U(X, H(X, Y_{M_t}))$ instead. Additionally, the logic behind the proof of Proposition 1 is unclear, particularly why $E_{x \in \mathcal{X}}(U(Y_{M_{t+1})}=0)$ holds. And Observation 2 is also confusing. Why should the absolute difference in distance measures equate to the difference in utilities?
3.	The authors devote substantial space to describing the experiment and results related to monetary incentives. However, since these are empirical observations of a single confounding factor in a specific scenario, they provide limited insight and generalizability from a broader perspective

**Questions:**

All my questions are listed in the weakness part.

---

> ### Author Response · Authors · 2024-11-22
>
> We thank the reviewer for pointing out the relevance of this research topic and for their feedback.
>
> 1) Our contribution focuses not on introducing new methodologies, but on describing the phenomenon in a way that informs the thinking around algorithmic deployment in a collaborative setting. By introducing the latent utility, we aim to provide a foundation for characterizing deployment strategies that may be generally beneficial or harmful under certain assumptions in the collaborative setting. While we agree that addressing the implied methodological challenges are an important direction for future work, our goal is to offer a framework within which such challenges can be explored and discussed.
>
> 2) We thank the reviewer for pointing out this inconsistency in our notation. We will denote $\delta_{M_t}^X$ as the delta determined by Mt and $X$ and $\delta_{M_t}$ for the expected delta over all $X$ and all $H$.
> Thank you for pointing out the missing $X$ in Definition 3. We changed that in our revised version of the paper along with your other suggestions.
>
> 3) We think there is a misunderstanding of our notation for the proof of proposition 1. The expression $L(Y_{M_{t+1}}, Y_{H_t})=0$ is an annotation to the equality. The loss is 0 because of our perfect learning assumption. This is not a statement about $Ex\in X(U(Y_{M_{t+1}}))$, ie, it is not assumed to be equal to zero. Observation 2 follows from the definition of a utility function (Definition 1) which requires that utility is a proximity measure. However, we take your comment as an opportunity to describe the properties used for the proof of proposition 1 in more detail in the appendix.
>
> 4) Two other reviewers in the team have emphasized that understanding incentives and human preferences when making decisions, with or without algorithmic recommendations, is important to understand the shape of the collaborative characteristic function. We are trying to strike the right balance between these two opposite views (shorten it vs. emphasize it). We faced the trade-off of exploring more factors affecting the collaborative characteristic function but with much less statistical power. We decided to investigate monetary incentives with high statistical power (which turned out to be important, given the null result). We see that as one example through which collaborative characteristic functions (which we hope are the contribution of our work) can vary. We hope this opens up several works on exploring collaborative characteristic functions in the real world, as we emphasize in the conclusions.

---

### Official Review · Reviewer_DvUe · 2024-11-03

**Soundness:** 2
**Presentation:** 2
**Contribution:** 2
**Rating:** 5
**Confidence:** 2

**Summary:**

This paper primarily presents a new dynamic framework for thinking about the deployment of ML models in performative human-ML collaborative systems, helping to understand how ML influences human decision-making processes. This research is intriguing and has practical value.

**Strengths:**

This paper has several strengths, as follows:
1.	A new dynamic framework is proposed for considering the deployment of ML models in human-ML collaborative systems.
2.	The involvement of participants in real-world scenarios enhances the credibility of the research. The design of the empirical study allows for clear identification of the actual ground truth, providing evidence for the research results.
3.	The findings of the paper have practical significance, aiding companies in optimizing the training and deployment strategies of ML models.

**Weaknesses:**

1.	The paper is hard to follow, the complexity may make it difficult for readers to understand.
2.	The research focuses primarily on the knapsack problem scenario, which may limit the generalizability of the results. It is recommended that the authors consider validation in different types of problems to enhance applicability.
3.	The paper mentions the failure to find a positive impact of incentive mechanisms on human decision quality, and the explanation for this phenomenon is insufficient, leading to a superficial discussion of the incentive mechanisms without exploring their potential reasons.

**Questions:**

1.	The paper initially presents that current human-ML collaborative systems face three crucial challenges, but the subsequent text does not detail the innovations made in addressing or alleviates these three issues. I hope to see a clear exploration of how the paper addresses or alleviates each of these challenges in the introduction.
2.	A deeper discussion on incentive mechanisms: Provide more discussion on the ineffectiveness of incentive mechanisms to help readers understand the potential reasons of this phenomenon.
3.	The contributions are trivial, making readers difficult to understand the key points of this paper. I hope the author can rewrite their contributions.
4.	In Definition 1, the definitions of Ymin and Y' are not specified. In Defination 5, the definition of x1…xn  should be placed in the main text.

---

> ### Author Response · Authors · 2024-11-22
> **Rebuttal of weaknesses**
>
> We thank the reviewer for highlighting the strengths of our papers. The reviewer is right that the paper already has a lot of content (and may be complex as a result), which is why we do not go deep into some of the extensions that the reviewer mentions (such as alternative problems to the knapsack, or exploring in depth why monetary incentives do not work). We tried our best to make the description as clear as possible. We will put substantial effort into improving the writing for a camera-ready version (and have already adjusted some wording as per the reviewer’s more specific questions).
>
> We also agree that the knapsack problem is just one problem. However, its advantages make the application affordable with humans recruited through Prolific. In particular: participants require little training to be able to solve it (making the recruitment of study participants affordable), the optimal solution is not obvious to humans but easy for us to calculate (so we can compare solutions to the undisputable ground truth), the optimal knapsack value is unique and unambiguous, even if there may exist more than one optimal solution (much harder to say for images and text), and we can generate instances at almost 0 cost. While we are aiming at designing user studies with more common learning tasks for future work, we hope that one empirical context (with variations in monetary incentives) is worthwhile as a first empirical setting. In the conclusions, we emphasize the importance of additional and realistic empirical settings as follow-up research: *“we see this as a first proof of concept of collaborative characteristic functions, but much more work is needed to estimate these functions in real-world settings.”*
>
> We agree that an in-depth analysis of monetary incentives would be worthwhile. In practice, in our setting, understanding why humans react to monetary incentives is important to understand changes in the shape of our collaborative characteristic function and corresponding learning paths (from our work, we know at least that the lack of an effect of monetary incentives does not come from not understanding them, since we tested them with and without a comprehension question). However, we don’t see that as the main objective of our paper. We acknowledge that the collaborative characteristic function can take any value, and our contribution lies in offering a framework to understand dynamic learning: *“The function ∆U can take any arbitrary form. Several factors can affect ∆U, e.g., humans’ attitudes towards algorithms, ML explanations, and monetary incentives (we empirically explore the latter in Section 4).”* In the conclusions as mentioned above, we emphasize that much more work is needed to understand the shape of these collaborative characteristic functions in the real world, and how humans’ incentives and preferences influence them.

---

> ### Author Response · Authors · 2024-11-22
> **Answers to questions**
>
> As for the specific questions:
>
> 1) We understand that the use of “challenges” can be misleading, and apologize for that. We are not trying to solve the three specific “challenges.” Rather, the three challenges characterize the context we study (make it interesting), and motivate our dynamic learning framework. We have rephrased that sentence accordingly:  *“Three key features characterize contexts where companies implement human-ML collaborative systems: 1) ML models learn from past human decisions, which are often only an approximation to the ground truth (noisy labels); 2) ML models are rolled out to help future human decisions, affecting the data-generating process of human-ML collaboration that then influences future updates to ML models (performative predictions as in Perdomo (2020)); and 3) the quality of the human-ML collaborative prediction of the ground truth may change as a function of incentives and other human factors. These features create a dynamic learning process.”*
>
> 2) We are interested in incentive mechanisms only to the extent that they change our collaborative characteristic function (and other reviewers have also emphasized that this is only one mechanism through which collaborative characteristic functions can change shape). We are trying to strike the right balance between these two opposite views. We discuss at least one result on incentives in the paragraph starting with *“The null effect of monetary incentives is not due to the fact that users did not understand the bonus structure…”* We also now have added text to emphasize that the discussion of incentive mechanisms is beyond the scope of our paper: *“While a deeper exploration of incentive mechanisms is beyond the scope of this paper, future research should explore how incentive design can change the shape of collaborative characteristic functions. We return to this in the Conclusions.”* Finally, our conclusions already emphasize that more work is needed along this dimension: *“Studying the interaction of monetary incentives and ML performance is an important extension. The null result of monetary incentives should be interpreted within our context. Specifically, the study participants received payments above minimum wage, and we only tested different levels of linear performance bonuses. It would be valuable to extend our work to evaluate the extent to which alternative base payments or non-linear bonuses may induce different levels of quality and effort by participants and thus collaborative characteristic functions of varying shapes.”*
>
> 3) Our contributions highlight the theoretical framework, the empirical proof of concept, and the practical implications for companies deploying recommendations to help humans’ decision-making. We hope some of the clarifications above and rewriting of the text have helped clarify our work.
> In Definition 1, we specify $Y_{min}$​ and $Y′$ as elements of $\mathcal{Y}$. $Y_{min}$ is subsequently used in the first property to characterize the minimum utility for a given $X$, while $Y′$ is utilized in other points of Definition 1. We make sure this is explicit in the new version of the paper. We also appreciate your suggestion regarding the placement of $x_1, …, x_n$​ in Definition 5. In the revised version of our paper, we have incorporated this definition into the main text as per your recommendation.

---

### Official Review · Reviewer_Zknp · 2024-11-03

**Soundness:** 3
**Presentation:** 2
**Contribution:** 3
**Rating:** 5
**Confidence:** 3

**Summary:**

This paper studies the dynamic model of performative human-ml collaboration from both theoretical and empirical perspectives. The paper introduces the notion, the Collaborative Characteristic Function which connect the predicted label and the unknown ground truth. The paper does some empirical study that involves real human on the knapsack problems. Experimental results show that human tend to improve the model's performance, and human may submit worse results than the prediction by models.

**Strengths:**

Originality: A substantive assessment of the strengths of the paper, touching on each of the following dimensions: originality, quality, clarity, and significance.
Quality: The paper is supported by a robust empirical study involving 1,408 participants working on the knapsack problem. The statistical analyses performed provide strong support for the conclusions drawn, particularly regarding human improvements over ML predictions. Additionally, the paper critically examines the impact of monetary incentives on decision quality, contributing valuable insights to the field.
Clarity: The paper's motivation and conclusion are clear.
Significance: The paper gives some suggestions about  the consideration of human behavior and the selection of the dataset to train the model.

**Weaknesses:**

1.Abstractness of Problem Domains: The study does not focus on specific classification or regression tasks, which makes the findings somewhat abstract. A more concrete application would enhance the practical relevance of the research.
2.Limited Application Scope: The research primarily concentrates on the knapsack problem, neglecting more realistic scenarios, such as medical diagnosis. Exploring applications in critical areas like healthcare would significantly increase the paper's impact and relevance.
3.Participant Preference Variability: While the study involves 1,408 participants, it lacks a detailed analysis of their preference differences. Understanding how individual preferences might affect decision-making is essential, as these variations could lead to suboptimal choices in certain instances.
4.Simulation of Human Behavior: Beyond conducting real experiments with participants, the paper does not explore the potential for simulating human behavior. Employing simulations could reduce the costs associated with extensive human experimentation while still providing valuable insights into collaborative decision-making dynamics.

**Questions:**

Can the author open-source the dataset provided by real human?

---

> ### Author Response · Authors · 2024-11-22
>
> We thank the reviewer for their insightful comments and the question.
>
> We will publish the human-labeled data along with the prediction of all ML models for each knapsack instance. We hope that this will be a valuable dataset for other research areas such as learning-to-defer, which usually have only few datasets with human and ML labels.
> We will also release the code for generating the hard knapsack instances, the models and all of our code for the data analysis, plot and post-processing of the instances. We will also release the code for our study platform which can be easily adapted for other experiments.
>
> As for the reviewer’s concerns, we agree that the knapsack problem is abstract. We believe that, as a first proof of concept, the advantages of the knapsack problems outweigh the limitations that the reviewer correctly pointed out. In particular: participants require little training to be able to solve it (making the recruitment of study participants affordable), the optimal solution is not obvious to humans but easy for us to calculate (so we can compare solutions to the undisputable ground truth), the optimal knapsack value is unique and unambiguous, even if there may exist more than one optimal solution (much harder to say for images and text), and we can generate instances at almost 0 cost. While we are aiming at designing user studies with more common learning tasks for future work, we hope that the benefits outweigh the disadvantages, as a first empirical setting. In the conclusions, we emphasize the importance of realistic empirical settings as follow-up research: *“we see this as a first proof of concept of collaborative characteristic functions, but much more work is needed to estimate these functions in real-world settings.”*
>
> We agree that an in-depth analysis of human decisions and why or why not humans follow algorithmic recommendations is important. In practice, in our setting, understanding why humans react to algorithmic recommendations the way they do or why humans make the decisions they make effectively changes the shape of our collaborative characteristic function and corresponding learning paths. In the paper, we already have one empirical example that could potentially change humans’ decisions (monetary incentives), but exploring more of them would be a paper in and of itself. We emphasize this in the paper when we present the collaborative characteristic function (slightly edited from the previous version to incorporate the reviewer’s feedback): *“The function ∆U can take any arbitrary form. Several factors can affect ∆U, e.g., humans’ attitudes towards algorithms, ML explanations, and monetary incentives (we empirically explore the latter in Section 4).”*
> We hope that making the data publicly available would allow others to dive deeper into this important question.
>
> We agree that simulations would have been cheaper and faster. We feared that the criticism at that point would have been the lack of empirical data backing our framework. For example, the review team emphasizes the involvement of study participants in real-world scenarios as important to increase the credibility of the work. However, we can certainly add a simulation (of both a learning path that leads to a good equilibrium and a learning path that leads to a bad equilibrium) in the appendix for a camera-ready version of the paper.

---

### Official Review · Reviewer_kfdp · 2024-11-03

**Soundness:** 3
**Presentation:** 2
**Contribution:** 1
**Rating:** 3
**Confidence:** 3

**Summary:**

* This paper examines Human-ML collaboration under performative prediction settings through theoretical analysis and an empirical experiment on ML-assisted combinatorial knapsack problems.
* In the setup, users interact with a predictive system in discrete time steps. At each time step $t$, a model $M_t$ predicts a label $Y_{M_t}$ based on features $X$. This prediction serves as decision support for a human decision-maker, who then makes their own prediction $Y_{H_t}$​​. Pairs $(X,Y_{H_t})$ are used to train the subsequent model $M_{t+1}$, and it is assumed that $M_{t+1}$​ perfectly aligns with its training distribution. Definition 1 introduces utility $\mathbb{U}(X, Y)$ for prediction-label pairs, defining its properties axiomatically. It then defines a the collaborative characteristic function which captures one-step utility improvement, and $\\mathbb{L}_{\\Delta\\mathbb{U}}(s,t)$ as the trajectory of expected utilities for a system whose initial utility is $s$. Propositions 1 and 2 show that utility trajectories converge under monotonicity assumptions.
* The empirical section evaluates the impact of model-based advice on human solutions for the 0-1 knapsack problem. Human participants interact with an ML-supported system to solve knapsack problems, possibly receiving predictions of the optimal solution. Six models with varying accuracy were trained before the experiment using synthetic optimal solutions, and each experimental group received distinct models and possibly different monetary incentives. Results indicate that incentivization schemes had no significant impact on solution quality, while decision-support quality correlated with human solution quality. Collaborative learning trajectories were presented based on these results.

**Strengths:**

* The paper addresses a well-motivated topic.
* The empirical analysis is grounded in data from real human subjects.
* Results seem to provide interesting insights into ML-assisted decision-making contexts.

**Weaknesses:**

* The paper claims to provide an empirical evaluation of performative prediction but seems to lack essential elements of this setup. Specifically, prediction models were trained on synthetic data before the experiment, and the experiment does not include "feedback loops" which are a defining component of performative prediction.
* The theoretical analysis applies to a limited form of performative prediction, assuming that utility trajectories are determined solely by population-wide average utility, without taking the structure of the predictor into account. Functions like the collaborative learning path are interesting, it is not clear whether the definition are applicable in more general scenarios.
* The empirical approach uses an atypical learning task: predicting a binary solution vector for a combinatorial 0-1 knapsack problem based on random synthetic instances and optimal solutions. The paper notes a possible analogy to multi-task classification, but it’s unclear how results extend to conventional ML tasks on non-synthetic data.

**Questions:**

* When are the conditions in Definition 1 expected to hold? Examples of suitable utility functions in binary classification and scalar regression can be very helpful.
* Could notation in eq. (1) be clarified? Specifically, $Y_{H_{t-1}}$ seems to appear both as an argument of the function, and as a variable sampled from $D_{t-1}$.
* How was Appendix Figure 6 (L403) generated?
* In the theoretical analysis, how would results change if the training set in each step was finite?

---

> ### Author Response · Authors · 2024-11-22
> **Rebuttal of weaknesses**
>
> We thank the reviewer for their extensive summary, comments, and questions. The reviewer is right that our empirical study is just an approximation of the performative prediction. We take care in calling it *“a proof of concept”* in the abstract and many other parts of the paper. In the contributions, we have adjusted our wording to further emphasize this: *“As a proof of concept for our theory, we provide some empirical insights in a context where humans solve knapsack problems with the help of machine learning predictions.”* We emphasize it as a limitation in the conclusions (second to last paragraph). Yet, there are several reason for this simplification: a simulation of the iterative process, which is easier to implement, would have lacked real human feedback; a study with the full iterative process (where we train an algorithm, deploy it, collect data from humans, train again, and so on) was too costly to implement, and risked suffering from changes in the compositions of study participants solving the problems.
>
> The advantage of our simplification is that it provides a first approximation to a phenomenon that we think is very important and (with generative AI) will become increasingly prevalent: subsequent model deployments affect the data generating process of data inputs to future models in ways that can deviate from the undisputable ground truth. We hope this is enough of a contribution, while acknowledging the limitations that the referee correctly points out. For example, in the conclusions we highlight that *“We see this as a first proof of concept of collaborative characteristic functions, but much more work is needed to estimate these functions in real-world settings.”*
>
> We do not see our assumption that utility trajectories are determined solely by population-wide average utility as too strict. In practice, assuming that the firm learning process perfectly learns the human-ML solution in the previous iteration allows us to move horizontally from the collaborative characteristic function to the 45-degree line (Figure 1). Deviations from this assumption would imply that the mapping from the human-ML performance to the next ML performance need not lie on the 45-degree line. As long as they are monotonically increasing (higher human-ML performance leads to higher next round’s ML performance), our insights remain.
>
> We also agree that the 0-1 Knapsack problem is not a standard learning task. However the knapsack problem has many desirable properties for the empirical investigation of human-ML collaboration: participants require little training to be able to solve it (making recruitment of study participants affordable), the optimal solution is not obvious to humans but easy for us to calculate (so we can compare solutions to the undisputable ground truth), the optimal knapsack value is unique and unambiguous, even if there may exist more than one optimal solution (much harder to say for images and text), and we can generate instances at almost 0 cost. While we are aiming at designing user studies with more common learning tasks for future work, we hope that the benefits outweigh the disadvantages, as a first empirical setting.

---

> ### Author Response · Authors · 2024-11-22
> **Answer to Questions**
>
> Regarding the reviewer’s specific questions:
>
> We provide two utility functions in the paper for the 0-1 knapsack problem modeled as a multilabel binary classification problem for which Definition 1 holds. For one dimensional binary classification, the inverted 0-1 loss would satisfy all criteria of definition 1. It is bounded, epsilon sensitive because there are only two possible labels, which also makes the distance measure obsolete. For scalar regression, any quantized (eps sensitivity), bounded and inverted (low distance $\Leftrightarrow$ high utility) distance metric should satisfy this definition. The main goal of the definition is to rule out two things: 1) that we will never reach a stable point because even the smallest improvement will be learned (epsilon sensitivity). This is automatically true in any real world setting. 2) That we don’t reach stable points because we move towards infinity utility forever. We would argue that this is also satisfied in any real world setting.
> Thank you for the remarks about eq. 1, which we will elaborate on in a potential camera ready version. The idea was to get the expectation over humans (outer expectation) and within humans over all instances that a single human works on (inner expectation). We see that this can be confusing as our notation suggests that this is a specific solution.
>
> Appendix Figure 6 was generated by sampling 10,25,50,75,100% of the human data (without the help of ml) that we collected during our experiment. We then trained the same model architecture on that data as the one in our study. We resampled+retrained 500 times for each sample size (10,25,50,75,100%). The goal was to check whether we exposed the humans to reasonable model performances when we trained on synthetic data (reasonable in the sense that the model performance achieved with human labeled  data is similar to that of models trained on synthetic data).
>
> Thank you for the question of what would happen if we would deal with finite data in our theory as this is an interesting one that we are currently working on as an extension of this work. We see it as very related to your prior comment that “utility trajectories are determined solely by population-wide average utility.” Deviations arising from finite data would imply that the mapping from the human-ML performance to the next ML performance need not lie on the 45-degree line (and may display variance that is a function of the sample size). As long as they are monotonically increasing (higher human-ML performance leads to higher next round’s ML performance), our insights remain, but the speed of convergence to equilibrium may be affected. This direction has important deployment decision implications and we think that the framework presented in this paper invites to explore various new related directions.

---

### Meta-Review · Area_Chair_kbuF · 2024-12-09

**Metareview:**

The paper presents a dynamic framework for performative human-ML collaboration, modeling how human decisions influenced by ML predictions can alter the data-generating process. The authors propose a utility-based theoretical model using collaborative characteristic functions to describe human-ML interactions. Empirical evaluations involve participants solving combinatorial knapsack problems with varying levels of ML assistance.

The main strengths raised by the reviewers include the problem relevance and combination of theoretical modeling and real-world user experiments. However, several concerns were raised, including limited generalizability due to the focus on the knapsack problem and reliance on synthetic data.  Overall, the paper would benefit from another round of revisions and review. We hope the authors find the reviewer comments helpful.

**Additional Comments On Reviewer Discussion:**

There is a consensus among the reviewers.

---

### Decision · Program_Chairs · 2025-01-22

Reject